# EXPECTED RETURN SYMMETRIES

**Darius Muglich**[*]
University of Oxford
darius@robots.ox.ac.uk

**Johannes Forkel**[*]
University of Oxford
johannes.forkel@eng.ox.ac.uk

**Elise van der Pol**
Microsoft Research AI for Science
evanderpol@microsoft.com

**Jakob Foerster**
University of Oxford
jakob.foerster@eng.ox.ac.uk

## ABSTRACT

Symmetry is an important inductive bias that can improve model robustness and generalization across many deep learning domains. In multi-agent settings, *a priori known* symmetries have been shown to address a fundamental coordination failure mode known as *mutually incompatible symmetry breaking*; e.g. in a game where two independent agents can choose to move "left" or "right", and where a reward of $+1$ or $-1$ is received when the agents choose the same action or different actions, respectively. However, the efficient and automatic *discovery* of environment symmetries, in particular for decentralized partially observable Markov decision processes, remains an open problem. Furthermore, environmental symmetry breaking constitutes only one type of coordination failure, which motivates the search for a more accessible and broader symmetry class. In this paper, we introduce such a broader group of previously unexplored symmetries, which we call *expected return symmetries*, which contains environment symmetries as a subgroup. We show that agents trained to be compatible under the group of expected return symmetries achieve better zero-shot coordination results than those using environment symmetries. As an additional benefit, our method makes minimal a priori assumptions about the structure of their environment and does not require access to ground truth symmetries.

## 1 INTRODUCTION

Incorporating the symmetries of an underlying problem into models has had demonstrable success in improving generalization and accuracy across many different machine learning domains (Bronstein et al., 2021; Krizhevsky et al., 2017; Cohen et al., 2019; Finzi et al., 2020; Van der Pol et al., 2020). As an important example, using data augmentations and equivariant networks has been shown to improve *zero-shot coordination* (ZSC), the ability of independently trained agents to coordinate in cooperative multi-agent settings at test time (Hu et al., 2020; Muglich et al., 2022a). Without accounting for symmetries, independently trained agents can converge onto equivalent yet mutually incompatible policies during training, leading to coordination failure at test time. For example, one team of agents might use the color "blue" to signal "play" in a cooperative card game like Hanabi, while a different team might use the equivalent color "red". This issue, known as *mutually incompatible symmetry breaking*, can be mitigated by incorporating symmetries like color into the training process, as is done in other-play (OP) (Hu et al., 2020). OP addresses this by applying an independently sampled symmetry transformation to each agent during every training episode, ensuring compatibility amongst equivalent policies.

Environmental symmetry breaking is a form of over-coordination, where agents adopt arbitrary conventions that hinder new partners' adaptation within an episode. Yet, this is only one failure mode; even without non-trivial symmetries, agents may over-coordinate by forming overly specific conventions (see Example 2). Moreover, current symmetry-based methods assume *a priori* access to these symmetries, but automatically discovering them—especially in large-scale Dec-POMDPs (Oliehoek et al., 2007)—can be computationally infeasible (Narayanamurthy & Ravindran, 2008).

---

[*]Equal contribution

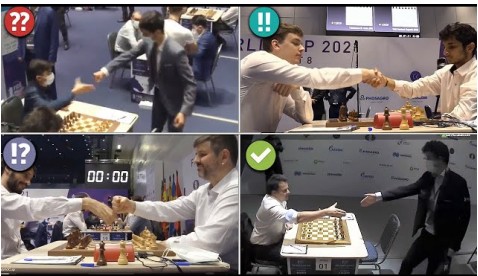

Figure 1: Mutually incompatible symmetry breaking between chess players is shown in the left side panels (Chess.com, 2021). Let $\pi_1$, $\pi_2$ represent the joint policies under which both players choose handshake, fist bump, respectively, and let the reward be $\pm 1$, depending on whether they match or not. The self-play score of both joint policies is 1, but the cross-play score between them is $-1$. Thus, policies incompatibly break the symmetry between handshake and fist bump.

To address these two issues, in this paper, we define the group of *expected return symmetries* (ER symmetries), a relatively underexplored symmetry group that contains the environment symmetries of a Dec-POMDP as a subgroup. Since in most cases ER symmetries are a strict superset of the environment symmetries, using them as the symmetry group for OP enforces training time compatibility with a greater number of policies.

We also introduce a scalable method for discovering approximate ER symmetries, which leverages gradient-based optimization to search for transformations that preserve expected returns across optimal policies. Furthermore, we show that ER symmetries better improve zero-shot coordination amongst independently trained agents than Dec-POMDP symmetries, while maintaining completely model-free assumptions. To summarize, our main contributions are:

1. We define the group of expected return symmetries and introduce novel algorithms for learning them.

2. We demonstrate that, when combined with the OP learning rule, expected return symmetries are significantly more effective at preventing over-coordination than Dec-POMDP symmetries. Furthermore, we demonstrate that our method is applicable in settings where both off-belief learning Hu et al. (2021) and cognitive hierarchies Cui et al. (2021) fail.

3. To the best of our knowledge, our method is the first symmetry-based method to improve zero-shot coordination without a priori/privileged environment information, such as of symmetries or dynamics.

## 2 BACKGROUND

### 2.1 DECENTRALIZED PARTIALLY OBSERVABLE MARKOV DECISION PROCESSES

We model cooperative multi-agent tasks as a Decentralized POMDP (Dec-POMDP) Oliehoek et al. (2007), defined by the 9-tuple

$$(\mathcal{S}, n, \{\mathcal{A}^i\}_{i=1}^n, \{\mathcal{O}^i\}_{i=1}^n, \mathcal{T}, \mathcal{R}, \{\mathcal{U}^i\}_{i=1}^n, H, \gamma).$$

Here, $\mathcal{S}$ is the state space, $n$ the number of agents, and for each agent $i$, $\mathcal{A}^i$ and $\mathcal{O}^i$ denote the local action and observation spaces (with joint action and observation spaces $\mathcal{A} = \times_{i=1}^n \mathcal{A}^i$ and $\mathcal{O} = \times_{i=1}^n \mathcal{O}^i$). The dynamics are given by

$$s_{t+1} \sim \mathcal{T}(s_{t+1} \mid s_t, a_t), \quad o_{t+1}^i \sim \mathcal{U}^i(o_{t+1}^i \mid s_{t+1}, a_t),$$

with rewards $r_{t+1} = \mathcal{R}(s_{t+1}, a_t)$, horizon $H$ (with terminal state $s_H$), and discount factor $\gamma$. Each agent $i$ follows a local policy $\pi^i$, which chooses local actions based on the local action-observation history (AOH) $\tau_t^i = (a_0^i, o_1^i, \ldots, a_{t-1}^i, o_t^i)$, while the joint policy $\pi = (\pi^1, \ldots, \pi^n)$ selects joint actions $a_t = (a_t^1, \ldots, a_t^n)$ based on the joint AOH $\tau_t = (\tau_t^1, ..., \tau_t^n)$, with probability

$$\pi(a_t|\tau_t) = \prod_{i=1}^n \pi^i(a_t^i|\tau_t^i).$$

The self-play objective is defined by the expected discounted return:

$$J(\pi) = \mathbb{E}_\pi \left[ \sum_{t=0}^{H-1} \gamma^t r_{t+1} \right].$$

## 2.2 ZERO-SHOT COORDINATION

The self-play objective is widely used in multi-agent reinforcement learning (MARL) (Samuel, 1959; Tesauro et al., 1995), where agents train together under a joint policy. While effective for coordination, it often results in arbitrary conventions that only work among agents trained together. However, many real-world tasks require coordination with unknown partners (Mariani et al., 2021; Resnick et al., 2018; Kakish et al., 2019), making this limitation problematic.

To address this, Hu et al. (2020) introduced zero-shot coordination (ZSC). In ZSC, agents first agree on a learning rule, which they each implement independently (e.g., cannot agree on seeds). Each agent then trains a joint policy in a Dec-POMDP environment, without communication or coordination between agents during training. Finally, agents participate in *cross-play* (XP), where joint policies trained by different agents are combined to evaluate the XP objective (defined here only for $n = 2$, but it can be extended to $n > 2$):

$$\text{XP}(\pi_1, \pi_2) := \frac{1}{2} \left( J(\pi_1^1, \pi_2^2) + J(\pi_2^1, \pi_1^2) \right), \tag{1}$$

for independently learned joint policies $\pi_1$ and $\pi_2$. ZSC aims for learning rules that optimize cross-play with other rational partners using the same minimal set of assumptions (e.g., no access to behavioral data or coordination experience with specific groups of agents). ZSC presents a promising approach to addressing real-world coordination challenges where relying on arbitrary conventions is impractical. ZSC has become a key benchmark for human-AI coordination and is an important step towards more generalized coordination capabilities (Ji et al., 2023; Hu et al., 2021).

## 2.3 SYMMETRY GROUPS AND OTHER-PLAY IN DEC-POMDPS

We consider symmetries that can be expressed as maps $\phi = (\phi_{\mathcal{S}}, \phi_{\mathcal{A}}, \phi_{\mathcal{O}})$, consisting of bijective maps $\phi_{\mathcal{S}} : \mathcal{S} \rightarrow \mathcal{S}$, $\phi_{\mathcal{A}^i} : \mathcal{A}^i \rightarrow \mathcal{A}^i$, and $\phi_{\mathcal{O}^i} : \mathcal{O}^i \rightarrow \mathcal{O}^i$, $i = 1, ..., n$, in the sense that $\phi_{\mathcal{A}}(a) = (\phi_{\mathcal{A}^1}(a^1), ..., \phi_{\mathcal{A}^n}(a^n))$, $\phi_{\mathcal{O}}(o) = (\phi_{\mathcal{O}^1}(o^1), ..., \phi_{\mathcal{O}^n}(o^n))$, for all $a \in \mathcal{A}$ and $o \in \mathcal{O}$. The set of all such maps forms a group, which we denote by $\Psi$. Given $\phi \in \Psi$, we slightly abuse notation and define $\phi(s) := \phi_{\mathcal{S}}(s)$, $\phi(a^i) := \phi_{\mathcal{A}^i}(a^i)$ and $\phi(o^i) := \phi_{\mathcal{O}^i}(o^i)$, for $s \in \mathcal{S}$, $a^i \in \mathcal{A}^i$ and $o^i \in \mathcal{O}^i$. Given a joint AOH $\tau_t = (\tau_t^1, ..., \tau_t^n)$, we define $\phi(\tau_t) := (\phi(\tau_t^1), ..., \phi(\tau_t^n))$, with $\phi(\tau_t^i) := (\phi(a_0^i), \phi(o_1^i), ..., \phi(a_{t-1}^i), \phi(o_t^i))$. Furthermore, we let a symmetry $\phi \in \Psi$ act on joint policies through the formula

$$\phi(\pi)^i(\phi(a_t^i) \mid \phi(\tau_t^i)) := \pi^i(a_t^i \mid \tau_t^i). \tag{2}$$

Any subgroup $\Phi \subset \Psi$ partitions the space of joint policies into disjoint equivalence classes: given a joint policy $\pi$, we define its equivalence class $[\pi] := \{\phi(\pi) : \phi \in \Phi\}$.

**Definition 1** (Dec-POMDP Symmetries). A map $\phi \in \Psi$ is called a Dec-POMDP symmetry if for all $(s_t, a_t, s_{t+1}, o_{t+1}) \in \mathcal{S} \times \mathcal{A} \times \mathcal{S} \times \mathcal{O}$ it holds that

$$\begin{aligned} \mathcal{T}(s_{t+1} \mid s_t, a_t) &= \mathcal{T}(\phi(s_{t+1}) \mid \phi(s_t), \phi(a_t)), \\ \mathcal{U}(o_{t+1} \mid s_{t+1}, a_t) &= \mathcal{U}(\phi(o_{t+1}) \mid, \phi(s_{t+1}), \phi(a_t)), \\ \mathcal{R}(s_{t+1}, a_t) &= \mathcal{R}(\phi(s_{t+1}), \phi(a_t)). \end{aligned} \tag{3}$$

We denote the set of all the Dec-POMDP symmetries of a given Dec-POMDP by $\Phi^{\text{MDP}}$.

Dec-POMDP symmetries form a subgroup of $\Psi$, which corresponds to relabelings of the state, action, and observation spaces that leave all the transition and reward functions of the Dec-POMDP unchanged (see Example 2). We note that Dec-POMDP symmetries can be extended to also include permutations between players (Treutlein et al., 2021).

Hu et al. (2020) demonstrate that policies equivalent under $\Phi^{\text{MDP}}$ are prone to mutually incompatible symmetry breaking: without biases like initialization or reward shaping, the learning rule may converge to either $\pi$ or its equivalent $\phi(\pi)$, as the learning process cannot distinguish between them due to the Dec-POMDP symmetries. Although $J(\pi) = J(\phi(\pi))$, the cross-play score may suffer, i.e. $J(\pi) > \text{XP}(\pi, \phi(\pi))$, meaning $\pi$ and $\phi(\pi)$ were not trained to be compatible. See Figure 1.

To constrain policies to be compatible with policies that are equivalent with respect to the symmetry group $\Phi^{\text{MDP}}$, Hu et al. (2020) introduced the *other-play objective* (can be extended to $n > 2$):

**Definition 2** (Other-Play (OP) Objective). Given a Dec-POMDP and a symmetry group $\Phi \subset \Psi$, we define the other-play (OP) objective $\text{OP}^\Phi : \Pi \to \mathbb{R}$ w.r.t. $\Phi$ by

$$\text{OP}^\Phi(\pi) := \mathbb{E}_{\tilde{\pi} \in [\pi]} \left[ \text{XP}(\pi, \tilde{\pi}) \right] = \frac{1}{2|[\pi]|} \sum_{\tilde{\pi} \in [\pi]} \left( J(\pi^1, \tilde{\pi}^2) + J(\tilde{\pi}^1, \pi^2) \right). \tag{4}$$

Hu et al. (2020) proposed the *OP learning rule* $\pi_* = \arg\sup_\pi \text{OP}^\Phi(\pi)$, for $\Phi = \Phi^{\text{MDP}}$. Agents trained using the OP objective take into account modes of symmetry breaking resulting from the fact that a test-time partner is unbiased in their choice between $\phi(\pi)$ and $\pi$, $\forall \phi \in \Phi$.

## 3 METHOD

In Section 3.1, we generalize the OP objective (Equation 4) to be defined over a general symmetry group $\Phi$ and highlight desirable properties that a symmetry group for OP should satisfy. In Section 3.2, we define the group of expected return symmetries, which we argue is better suited for OP than Dec-POMDP symmetries by way of the aforementioned desirable properties. In Section 3.3, we propose a method for learning expected return symmetries.

### 3.1 OTHER-PLAY OVER GENERAL SYMMETRY GROUPS

While Hu et al. (2020) introduced symmetry breaking w.r.t. $\Phi^{\text{MDP}}$, we extend the definition for a general symmetry group $\Phi$:

**Definition 3** (Symmetry Breaking). Given a symmetry group $\Phi$, a joint policy $\pi$ is said to incompatibly break symmetry w.r.t. $\phi \in \Phi$ if $J(\pi) > \text{XP}(\pi, \phi(\pi))$, and w.r.t. $\Phi$ if $J(\pi) > \text{OP}^\Phi(\pi)$.

The OP objective (Equation 4) evaluates the expected return when an agent from one policy is matched in a team with members of randomly chosen policies from the same equivalence class induced by $\Phi$. Thus OP optimal policies are maximally compatible with policies within their equivalence class, as they best avoid incompatible symmetry breaking w.r.t. $\Phi$. We note that different $\text{OP}^\Phi$-optimal policies are not necessarily in the same equivalence class and can therefore be incompatible; for example, when $\Phi = \textbf{Id}$, $\text{OP}^\Phi$ reduces to SP, and each $\text{OP}^\Phi$-optimal policy forms its own one-element equivalence class. We denote $\Pi_*^\Phi$ as the set of all optimal policies under $\text{OP}^\Phi$.

Rational and independent ZSC agents would therefore choose a symmetry group $\Phi$ such that:

1. **(Diversity within Equivalence Classes)** The choice of $\Phi$ should ensure that for all $\pi \in \Pi_*^\Phi$, the equivalence class $[\pi]$ is meaningfully *diverse*. This diversity should be such that using OP to enforce $\pi$ and $[\pi]$ to be compatible makes $\pi$ broadly compatible with policies it could encounter at test-time, assuming other agents also adopt $\text{OP}^\Phi$ as their learning rule.

2. **(Optimality within Equivalence Classes)** $\Phi$ should separate poor policies from good policies, i.e. $\text{OP}^\Phi(\pi') \approx \text{OP}^\Phi(\pi)$, for any $\pi \in \Pi_*^\Phi$ and $\pi' \in [\pi]$, since otherwise there is no reason to constrain oneself to be compatible with $\pi'$ (a rational test-time partner would not use $\pi'$).

**Example 1.** For $\Phi = \{\textbf{Id}\}$, $\text{OP}^\Phi = \text{SP}$, which corresponds to perfectly preserving optimality, but not introducing any diversity to the equivalence class; i.e. perfect satisfaction of Item 2 but extremely poor satisfaction of Item 1. On the other extreme, we can consider $\Phi$ to be the set of all bijections on $\Pi$, which enforces compatibility with every possible test-time partner (i.e., rational partners), but also with all other possible policies (which are mostly poor), thus converging to the best response to a random player; i.e., perfect satisfaction of Item 1 but extremely poor satisfaction of Item 2.

The goal of ZSC is to find a learning rule that maximizes the expected XP score between independently trained test-time partners. For the learning rule $\text{OP}^\Phi$ the expected XP score is:

$$\text{XP}(\text{OP}_*^\Phi) := \mathbb{E}_{\pi_1, \pi_2 \sim \Pi_*^\Phi} \left[ \text{XP}(\pi_1, \pi_2) \right]. \tag{5}$$

Thus one can interpret $\max_{\pi \in \Pi} \text{OP}^\Phi(\pi)$ as estimating $\text{XP}(\text{OP}_*^\Phi)$, but only through the cross-play scores *within* a given equivalence class $[\pi]$ induced by $\Phi$:

$$\max_{\pi \in \Pi} \text{OP}^\Phi(\pi) = \mathbb{E}_{\pi \sim \Pi_*^\Phi} \left[ \text{OP}^\Phi(\pi) \right] = \mathbb{E}_{\pi_1 \sim \Pi_*^\Phi} \left[ \mathbb{E}_{\pi_2 \sim [\pi_1]} \left[ \text{XP}(\pi_1, \pi_2) \right] \right]. \tag{6}$$

Since optimal $\text{OP}^\Phi$ policies are not trained to be compatible across *different* equivalence classes, we can always assume w.l.o.g. that $\text{OP}^\Phi(\pi) > \mathbb{E}_{\pi'' \in [\pi']}[\text{XP}(\pi, \pi'')]$ for any $\pi, \pi' \in \Pi_*^\Phi$ for which $\pi \notin [\pi']$ (else one just merges $[\pi]$ and $[\pi']$ by adding the transposition[1] between $\pi$ and $\pi'$ to $\Phi$). Thus if Item 2 is satisfied perfectly, it follows that

$$\max_{\pi \in \Pi} \text{OP}^\Phi(\pi) \geq \text{XP}(\text{OP}_*^\Phi), \tag{7}$$

with equality if and only if $[\pi] = \Pi_*^\Phi$ for any and thus all $\pi \in \Pi_*^\Phi$. This means that if one finds a symmetry group $\Phi$ which satisfies both of Items 1 and 2 perfectly, then $[\pi] = \Pi_*^\Phi$ for any $\pi \in \Pi_*^\Phi$ and $\text{OP}^\Phi(\pi) = \text{XP}(\text{OP}_*^\Phi)$. In other words, with such a choice of $\Phi$, agents during training account for any potential test-time partner produced by $\text{OP}^\Phi$, and only for such partners. Items 1 and 2 are thus desirable criteria for choosing a group $\Phi$ that makes $\text{OP}^\Phi$ a suitable learning rule for ZSC. However, these criteria alone are *not sufficient* for $\Phi$ to be optimal for ZSC, because agents in ZSC cannot choose a symmetry group $\Phi$ that is tailored to a specific Dec-POMDP. For example, suppose agents select $\Phi$ as the set of all bijections on $\Pi$ that leave a particular SP optimal policy $\pi$ unchanged. In this case, except for trivially simple Dec-POMDPs, we have $\Pi_*^\Phi = [\pi] = \{\pi\}$. This choice of $\Phi$ would trivially satisfy Item 1 (since $\{\pi\}$ is entirely representative of test-time policies) and Item 2 (since $\pi \in \Pi_*^\Phi$ and $\phi(\pi) = \pi, \forall \phi \in \Phi$). However, such a symmetry group is not permissible in ZSC because it is specifically constructed for a particular policy in a specific Dec-POMDP, violating the requirement for generality in ZSC. Therefore, while Items 1 and 2 are desirable properties, they are not sufficient on their own for choosing an appropriate symmetry group $\Phi$ for ZSC.

## 3.2 EXPECTED RETURN SYMMETRIES

We propose that the group $\Phi^{\text{ER}}$ of expected return (ER) symmetries, which can be learned with completely model-free assumptions, handles the above trade-off given by Items 1 and 2 favorably:

**Definition 4** (Expected Return Symmetries). For a fixed temperature $\alpha > 0$, define the set of Boltzmann exploratory policies as

$$\Pi^\alpha := \left\{ \pi \in \Pi \;\middle|\; \pi^i(a_t^i \mid \tau_t^i) = \frac{\exp\left(\frac{Q_\pi^i(\tau_t^i, a_t^i)}{\alpha}\right)}{\sum_{a^i \in \mathcal{A}^i(\tau_t^i)} \exp\left(\frac{Q_\pi^i(\tau_t^i, a^i)}{\alpha}\right)} \;,\, \forall a_t^i \in \mathcal{A}^i(\tau_t^i),\; \forall \tau_t^i,\; \forall i = 1, ..., n \right\},$$

where $Q_\pi^i(\tau_t^i, a_t^i)$ is the action-value function of $\pi^i$, when assuming that all other agents also follow $\pi$. The set of self-play optimal Boltzmann exploratory policies is then defined as

$$\Pi_*^\alpha := \arg\max_{\pi \in \Pi^\alpha} J(\pi).$$

Finally, define the group of expected return symmetries as

$$\Phi^{\text{ER}} := \left\{ \phi \in \Psi \;\middle|\; \forall \pi \in \Pi_*^\alpha : \phi(\pi) \in \Pi_*^\alpha \right\},$$

i.e., the subset of transformations preserving self-play optimality of Boltzmann-exploratory policies.

We will suppress the dependence of $\Phi^{\text{ER}}$ on $\alpha$, but it is important to define $\Phi^{\text{ER}}$ over self-play optimal Boltzmann-exploratory policies: if one were to allow self-play optimal policies that do not explore the entire space of AOHs in this structured way, then expected return symmetries would not be properly restricted in their effect on policies at suboptimal AOHs, and $\Phi^{\text{ER}}$ could contain symmetries that inhibit coordination (see Appendix C). For small $\alpha$, policies learned via Q-learning can be converted—albeit approximately, due to the algorithm being off-policy —into policies belonging to $\Pi^\alpha$ by using Boltzmann sampling on the learned local action values. As well, entropy-regularized policy gradient methods yield policies whose actor networks produce logits proportional to the corresponding Q-values, thereby resulting in policies in $\Pi^\alpha$ (see Appendix B.2 for details).

$\Phi^{\text{ER}}$ is a group under function composition (see Appendix B.1). Furthermore, since for any Dec-POMDP symmetry $\phi \in \Phi^{\text{MDP}}$ and any joint policy $\pi$ it holds that $J(\pi) = J(\phi(\pi))$, and also that $\phi(\pi)$ is Boltzmann exploratory, it follows that $\Phi^{\text{ER}}$ contains $\Phi^{\text{MDP}}$ as a subgroup (see Appendix B.1).

---

[1] A transposition is a permutation that swaps exactly two elements.

At a high level, $\Phi^{\mathrm{MDP}}$ captures coordination differences based only on relabeling actions and observations, offering limited diversity. In contrast, self-play-optimal policies can differ significantly in coordination strategies, beyond mere label permutations. By grouping such diverse policies into the same equivalence class, $\Phi^{\mathrm{ER}}$ better addresses Item 1 than $\Phi^{\mathrm{MDP}}$. While $\Phi^{\mathrm{ER}}$ may not fully satisfy Item 2 (as $\Phi^{\mathrm{ER}}$ is only required to preserve the expected return of self-play-optimal policies), we show in Section 4.4 that learned symmetries in $\Phi^{\mathrm{ER}}$ approximately meet this criterion. Overall, this suggests ZSC agents using $\mathrm{OP}^{\Phi^{\mathrm{ER}}}$ will coordinate better at test time than those using $\mathrm{OP}^{\Phi^{\mathrm{MDP}}}$. Section 4 confirms that $\Phi^{\mathrm{ER}}$ significantly improves (zero-shot) coordination across various environments.

The following example illustrates the advantage of ER symmetries over Dec-POMDP symmetries.

**Example 2.** Consider a cooperative communication game based on Hu et al. (2021). Alice observes a binary variable (pet) representing either "cat" or "dog". Her actions include turning a light bulb on for a reward of 0.01, light bulb off for a reward of 0, bailing for a reward of 1 which ends the game, or removing a barrier at a cost of 5 to let Bob directly observe the pet. Bob can bail for 0.5 or guess the pet, receiving 10 for a correct "cat" guess, 11 for a correct "dog" guess, and losing 10 for an incorrect guess.

In this game, there are two ways for Alice to communicate with Bob: a "cheap-talk" channel (turning the light on or off) and a "grounded" channel (removing the barrier). There are exactly two self-play optimal (Boltzmann exploratory) joint policies, and both use the cheap-talk channel with high probability: one policy associates "light on = dog, light off = cat," while the other assigns the reverse. These two policies are incompatible in cross-play however, where coordination fails if each agent follows a different encoding.

Due to the different rewards for cat/dog, this game contains no non-trivial Dec-POMDP symmetries, which means that OP with $\Phi^{\mathrm{MDP}}$ reduces to self-play, which, as just discussed, leads to coordination failure since independently trained agents will converge on either of the two self-play-optimal but cross-play-incompatible policies. Indeed, permuting "on/off" and "cat/dog" is not a Dec-POMDP symmetry because these environmental features have different reward dynamics.

However, these pairings *are* ER symmetries, as they transform the two self-play-optimal policies into one another and thus preserve their expected return. Therefore, these two policies are symmetric to each other w.r.t. $\Phi^{\mathrm{ER}}$, are put into the same equivalence class, and OP with $\Phi^{\mathrm{ER}}$ is able to anticipate their coordination failure. OP with $\Phi^{\mathrm{ER}}$ then chooses the optimal grounded policy, leading to the best possible cross-play score in this game for independent rational agents. This is demonstrated empirically in Section 4.2.

### 3.3 ALGORITHMIC APPROACHES

In practice, if $\Phi^{\mathrm{ER}}$ is large, we find that it can be effectively approximated by a limited number of learned ER symmetries, that are sufficiently diverse (see Section 3.1 for the importance of diversity). We therefore develop an algorithm to learn such a subset of ER symmetries, which we find in Section 4 is sufficient to significantly enhance zero-shot coordination across various environments.

Based on Equation 4, we formulate the following objective for learning ER symmetries:

$$\phi_{\theta^*} \quad \text{where} \quad \theta^* = \arg\inf_{\theta \in \Theta} \mathbb{E}_{\pi \sim \Pi'} \left[ \, |J(\pi) - J(\phi_\theta(\pi))| \, \right], \tag{8}$$

where $\{\phi_\theta : \theta \in \Theta\}$ is a parameterization, and $\Pi' \subset \Pi_*^\alpha$ is a fixed pool of SP optimal Boltzmann exploratory policies. The broader the set $\Pi'$ is, the more representative it will be of $\Pi_*^\alpha$, and hence the less likely the learned $\phi_{\theta^*}$ will overfit to a specific policy (i.e. not able to preserve expected return for other optimal policies outside the training set).

Since the policies in Equation 8 are approximately SP optimal, we can use the equivalent objective

$$\phi_{\theta^*} \quad \text{where} \quad \theta^* = \arg\sup_{\theta \in \Theta} \mathbb{E}_{\pi \sim \Pi'} \left[ J(\phi_\theta(\pi)) \right]. \tag{9}$$

Importantly, the optimization in Equation 9 focuses only on the ER symmetry $\phi_\theta$. In this process, we train the transformation $\phi_\theta$ within a reinforcement learning loop, but we keep the weights of the policies in $\Pi'$ fixed. See Algorithm 1 in Appendix E for details. The Boltzmann exploration of the policies in $\Pi'$ enforces that $\phi_\theta$ takes into account all possible AOHs during training, not just optimal ones. Furthermore, when training $\phi$ using policy gradient methods, we use entropy-regularization to ensure $\phi(\pi)$ yields a self-play optimal Boltzmann exploratory policy (see Appendix B.2 for details).

Recall that $\phi_\theta = \{\phi_{\mathcal{S},\theta}, \phi_{\mathcal{O},\theta}, \phi_{\mathcal{A},\theta}\}$. Since we are interested in expected return symmetries insofar as they act on the policy space rather than the Dec-POMDP itself, we fix $\phi_{\mathcal{S},\theta} = \mathbf{Id}$. Furthermore, since typically $|\mathcal{A}| \ll |\mathcal{O}|$, rather than learning both $\{\phi_{\mathcal{O},\theta}, \phi_{\mathcal{A},\theta}\}$ in a reinforcement learning loop, we can consider learning $\phi_{\mathcal{A},\theta}$ through search over tuples of permutations on the local action spaces; i.e. we initialize a fixed tuple of local action permutations as $\phi_{\mathcal{A},\theta}$ and learn $\phi_{\mathcal{O},\theta}$ as per Equation 9. See Algorithm 1 in Appendix E for an outline of this procedure.

Each local action space $\mathcal{A}^i$ allows for $\binom{|\mathcal{A}^i|}{2}$ distinct transpositions, representing the number of unique ways two actions can be permuted. Consequently, learning one observation transformation $\phi_{\mathcal{O},\theta}$ corresponding to every possible tuple of local action transpositions requires $O(\prod_{i=1}^n |\mathcal{A}^i|^2)$ optimizations of Equation 9 to perform an exhaustive search (i.e. $O(\prod_{i=1}^n |\mathcal{A}^i|^2)$ iterations of the outer for loop in Algorithm 1). However, in Section 4.4 we show that a non-exhaustive search that undersamples the space of tuples of transpositions is still sufficient for learning symmetries that significantly improve coordination amongst agents.

Note that the objective in Equation 9 does not enforce *closure under composition* (i.e. $\phi_1 \circ \phi_2 \in \Phi^{\mathrm{ER}}$ for all $\phi_1, \phi_2 \in \Phi^{\mathrm{ER}}$) or *invertibility* (i.e. for all $\phi \in \Phi^{\mathrm{ER}}$ there exists $\phi^{-1} \in \Phi^{\mathrm{ER}}$ such that $\phi^{-1} \circ \phi = \mathbf{Id}$), both necessary for $\Phi^{\mathrm{ER}}$ to form a group. To learn ER symmetries which are compositional and invertible, we use the following regularized objective:

$$\phi_{\theta^*} \ \text{where} \ \theta^* = \arg\max_{\theta \in \Theta} \mathbb{E}_{\pi \sim \Pi'}\left[(1-\lambda_1)J(\phi_\theta(\pi)) + \lambda_1 \cdot \mathbb{E}_{\hat{\phi}_i, \hat{\phi}_j \sim \hat{\Phi}^{\mathrm{ER}}}\left[J(\hat{\phi}_i \circ \phi_\theta \circ \hat{\phi}_j(\pi))\right]\right]$$
$$- \lambda_2 \mathbb{E}_{o \in \mathcal{O}}\left[d(o, \phi_\theta^2(o))^2\right] \quad (10)$$

where $d : \mathcal{O}^2 \to [0,\infty)$ is a metric, $\hat{\Phi}^{\mathrm{ER}}$ is a fixed pool of unregularized ER symmetries learned through Equation 9Algorithm 1, and $\lambda_1 \in [0,1)$, $\lambda_2 \in [0,\infty)$ control the regularization towards compositionality and invertibility, respectively. Since $\phi_{\mathcal{A},\theta}$ is a fixed transposition, $\phi_{\mathcal{A},\theta}^2 = \mathbf{Id}$ by design, so we can easily enforce $\phi_{\mathcal{O},\theta}^2 = \mathbf{Id}$. The objective can be optimized stochastically, to avoid computing multiple policy gradients per update. This is detailed in Algorithm 2 in Appendix E. Note that in the term $\mathbb{E}_{o \in \mathcal{O}}\left[d(o, \phi_\theta^2(o))^2\right]$ we abuse notation, and let $\phi_\theta$ map into a continuous extension of $\mathcal{O}$, otherwise this term would be locally constant with a gradient of zero almost everywhere.

We also propose an alternative objective for learning ER symmetries through XP maximization:

$$\phi_{\theta^*} \ \text{s.t.} \ \theta^* = \arg\sup_{\theta \in \Theta} \mathrm{XP}(\pi_i, \phi_\theta(\pi_j)), \quad (11)$$

where $\pi_i, \pi_j \in \Pi'$ are a pair of SP optimal policies chosen from the fixed training pool. If $\pi_i$ and $\pi_j$ belong to the same equivalence class induced by $\Phi^{\mathrm{ER}}$, then by definition there exists an ER symmetry $\phi$ that maximizes Equation 11 to the self-play optimum value of $J(\pi_i)$. Therefore, for each pair of optimal policies $\pi_i, \pi_j \in \Pi'$, we optimize Equation 11 over $\phi_\theta$, and save the $\phi_\theta$ that optimize Equation 11 to the highest value. We outline this approach in Algorithm 3 of Appendix E. We highlight a trade-off between the objectives of Equation 10 and Equation 11: while the former more directly optimizes for an ER symmetry, it assumes $\phi_{\mathcal{A},\theta}$ to be of a certain form, while the latter assumes no such form but tacitly assumes some pair in $\Pi'$ belong to the same equivalence class.

## 4 EXPERIMENTS

We evaluate our method in four different environments, focusing on how ER symmetries impact zero-shot coordination (ZSC) compared to self-play and other-play with Dec-POMDP symmetries. Specifically, we train independent agent populations that take advantage of ER symmetries and compare their cross-play performance within the population to baseline populations. The goal is to assess whether the use of ER symmetries leads to better coordination between agents than self-play or Dec-POMDP-symmetry-based training.

Populations of agents using ER symmetries for ZSC are formed as follows: each agent $i$ chooses $k$ seeds at random to train $k$ different optimal policies, $\Pi'_i$. Agent $i$ then independently performs ER symmetry discovery with their specific $\Pi'_i$ by optimizing Equation 9, Equation 10 or Equation 11, and among their learned transformations uses the $l$ that best preserve expected return as their ER symmetries. Agent $i$ then chooses $m$ seeds at random and uses their learned symmetries to train $m$ policies $\{\pi_{i,k}\}_{k=1}^m$ with reinforcement learning constrained by the learning rule in Equation 4; multiple policies ($m > 1$) are trained to mitigate the effect of a seed that sub-optimally explores the space. Agent $i$ then selects $\pi_i := \arg\max_{k=1,\ldots,m} J(\pi_{i,k})$, and deploys $\pi_i$ for cross-play.

Aside from the environments in Sections 4.1 and 4.2, we parameterize $\phi_{\mathcal{O},\theta}$ as a feed-forward neural network with two hidden layers. The experiments in Sections 4.3 and 4.4 use the JaxMARL environment and implementations (Rutherford et al., 2023). For details on our setup and hyperparameters, refer to Appendix A. See Appendix F for plots of interpretability of agent play. If accepted, we will release our full working code for all four environments.

## 4.1 ITERATED THREE-LEVER GAME

We consider a Dec-POMDP inspired by Treutlein et al. (2021) where two agents choose one of three levers simultaneously, earning $+1$ for a match and $-1$ otherwise over 2 rounds, with each agent observing the other's previous action. Thus, $|\mathcal{S}| = 1$, $\mathcal{A} = \mathcal{O} = \{1,2,3\} \times \{1,2,3\}$, $r_{t+1} = \mathbb{1}_{a_t^1 = a_t^2}$, and $o_{t+1}^1 = a_t^2$, $o_{t+1}^2 = a_t^1$.

There are 6 Dec-POMDP symmetries (the 6 permutations of levers). The optimal $\text{OP}^{\Phi^{\text{MDP}}}$ policy chooses a lever uniformly in round one; if the agents match, they repeat that choice, otherwise they switch to the unique unused lever. This yields an expected return of $4/3$ (optimal for ZSC given a $1/3$ chance of first-round success). Since $\Phi^{\text{ER}} = \Phi^{\text{MDP}}$, learning all of $\Phi^{\text{ER}}$ and applying $\text{OP}^{\Phi^{\text{ER}}}$ approximates the optimal ZSC policy.

Each ER symmetry agent first trains $k = 20$ SP optimal policies using IQL with shared Q-values, and then makes them Boltzmann exploratory with $\alpha = 1$. For each tuple of local action permutations, we select the tuple of local observation permutations that maximizes the objective in Equation (9) (averaged over 2000 episodes). The ERS agent then picks the $l = 6$ best ER symmetries. Each of 5 ERS agents learns exactly the 6 Dec-POMDP symmetries.

This game illustrates a setting where OP outperforms OBL (Hu et al., 2021)—OBL fails (in its turn-based form) as agents assuming uniform randomness converge to a uniform distribution. However, in the two-lever variant $\text{OP}^{\Phi^{\text{MDP}}}$ (and hence $\text{OP}^{\Phi^{\text{ER}}}$) fails due to the lack of a unique second-round choice, leaving repeat and switch policies in distinct, incompatible equivalence classes. This underscores a limitation in the symmetry group's expressivity.

## 4.2 CAT/DOG ENVIRONMENT

We take the cat/dog game from Example 2. We build a population of 5 independent Q-learning (IQL) (Tan, 1993) agents as a self-play baseline. We build a population of 5 ER symmetry agents, for which each agent trains $k = 10$ optimal self-play policies as $\Pi'$, and then optimizes Equation 9 with vanilla policy gradient (Sutton et al., 1999), where $\phi_{\mathcal{O},\theta}$ is parameterized as a probability distribution over all possible permutations of the observations (there being 2! for Alice and 4! for Bob). Equation 9 / Algorithm 1 suffices because permutations already satisfy compositionality and invertibility. Each ER symmetry agent uses $l = 3$ ER symmetries to then train $m = 3$ $\text{OP}^{\Phi^{\text{ER}}}$ policy.

We find ER symmetries can prevent over-coordination in settings where non-trivial Dec-POMDP symmetries do not even exist. It is worth mentioning that approaches based on cognitive hierarchies fail to find the optimal grounded communication protocol in this setting (Hu et al., 2021; Cui et al., 2021; Camerer et al., 2004), since they assume other agents follow random or lower-level strategies, and instead consistently converge onto "bailing" for a return of 1.

## 4.3 OVERCOOKED V2

Overcooked V2 is a recent AI benchmark for ZSC (Gessler et al., 2025), which improves on the cooperative multi-agent benchmark Overcooked (Carroll et al., 2019), by introducing asymmetric information and increased stochasticity, creating more nuanced coordination challenges. In this work, we concentrate on the "Grounded Coordination Simple" layout, leaving the exploration of additional layouts to future work.

For ZSC, we train a population of 5 IPPO (Yu et al., 2022) policies as a SP baseline, where each policy uses an RNN coupled with a CNN to process the observations. The population of ER symmetry agents each train $k = 12$ IPPO SP policies, to then use Equation 11 / Algorithm 3 to obtain $l = 16$ ER symmetries. Each agent trains $m = 3$ $\text{OP}^{\Phi^{\text{ER}}}$ policies using their learned symmetries.

The IPPO baseline shows a bimodal XP score distribution, with agents either highly compatible or incompatible. In Figure 2, the SP-optimal baseline scores a mean XP of 6.74, compared to 15.8

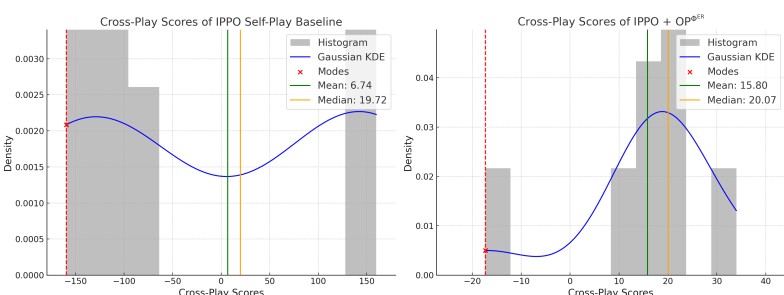

Figure 2: Cross-play score distribution of the IPPO self-play baseline population and the ER symmetry agent population in Overcooked V2. The baseline population achieves mean SP scores of $162.33 \pm 0.14$, and the ER symmetry population achieves mean SP scores of $27.81 \pm 0.3$.

for the $\mathrm{OP}^{\Phi^{ER}}$-optimal population. The ER symmetry population thus significantly reduces the SP-XP gap, leading to more consistent coordination. Overcooked V2's simultaneous moves further underscore the effectiveness of ER symmetries for ZSC.

## 4.4 HANABI

Hanabi (see Appendix G for details) is a challenging AI benchmark, and has served as the primary test bed for many algorithms designed for zero-shot coordination, ad-hoc teamplay, and other cooperative tasks (Bard et al., 2020; Cui et al., 2021; Nekoei et al., 2021; 2023; Muglich et al., 2022a;b).

**Preserves OP Optimality**

Since ER symmetries contain Dec-POMDP symmetries, and capture equivalences beyond just relabelling, they are clearly more diverse than Dec-POMDP symmetries, and hence better satisfy Item 1 from Section 3.1. We verify the ER symmetries also approximately satisfy Item 2.

We take the 11 learned regularized ER symmetries from above, denoting this set as $\hat{\Phi}^{ER}$. We find that $\mathbb{E}_{\pi \sim \Pi_*^{\hat{\Phi}^{ER}}} \left[ \mathrm{OP}^{\hat{\Phi}^{ER}}(\pi) \right] = 23.59 \pm 0.04$ and $\mathbb{E}_{\pi \sim \Pi_*^{\hat{\Phi}^{ER}}} \left[ \mathbb{E}_{\phi \sim \hat{\Phi}^{ER}} \left[ \mathrm{OP}^{\hat{\Phi}^{ER}}(\phi(\pi)) \right] \right] = 23.34 \pm 0.05$, where we train 5 $\mathrm{OP}^{\Phi^{ER}}$ policies. Thus, Item 2 is approximately satisfied by $\hat{\Phi}^{ER}$.

**Zero-Shot Coordination**

For ZSC we use two baselines: (1) a population of 5 IPPO agents, and (2) 5 IPPO agents with access to all Dec-POMDP symmetries constrained by the OP objective (Equation 4). We also train a population of ER symmetry agents that independently discover ER symmetries for the OP objective. Each ER agent uses $k = 6$ seeds to learn symmetries and saves the top $l = 11$ that best preserve expected return. Every population consists of 5 agents, where each agent trains $m = 5$ policies and deploys the one with the highest return. Symmetries are trained via Equation 10 by randomly selecting 64 fixed transpositions on the local action space, with the same transpositions fixed for both local policies to simplify the search in the symmetric game Hanabi.

Inspired by the symmetrizer in Treutlein et al. (2021); Muglich et al. (2022a); Van der Pol et al. (2020), we define the symmetrizer $S : \Pi \to \Pi$ by $S(\pi)(a_t \mid \tau_t) := \frac{1}{|[\pi]|} \sum_{\pi' \in [\pi]} \pi'(a_t \mid \tau_t)$, where $\Phi$ can be taken to be either $\Phi^{ER}$ or $\Phi^{MDP}$. Agents from any population can thus transform their policy with $S$ before deploying for cross-play, ensuring invariance of the deployed policy w.r.t. $\Phi$ (since $\phi(S(\pi)) = S(\pi), \forall \phi \in \Phi$). As per the empirical results below, the symmetrizer functions as a policy improvement operator for cross-play across multiple policy populations.

Table 1 shows that the agents using ER symmetries improve in ZSC over both baselines; this is even in spite of the Dec-POMDP agent population assuming access to environment symmetries. As well, even with just using a subset of transformations from $\hat{\Phi}^{ER}$, the return symmetry agents are able to converge on policies that generalize well in cross-play. We also notice that symmetrization with respect to $\Phi^{ER}$ or $\Phi^{MDP}$ improves coordination amongst agents of all populations considered; we can see that $\Phi^{ER}$ better improves the optimal self-play policy population, and $\Phi^{MDP}$ better improves the other two, which aligns with expectation since $\Phi^{ER}$ is only explicitly enforced to maintain invariance over optimal self-play policies, whereas $\Phi^{MDP}$ maintains invariance over any policy type.

Table 1: Self-play, within-population mean cross-play (XP) and median cross-play (XP(*)) scores are reported. The $OP^{\Phi^{MDP}}$ population used all 120 Dec-POMDP symmetries, whereas the $OP^{\Phi^{ER}}$ population used 11 ER symmetries. "MDP" indicates the population was symmetrized with Dec-POMDP symmetries at test time, and "ER" analogously indicates symmetrization with expected return symmetries. The ER symmetrizer uses 11 expected return symmetries.

| Model | Self-Play | XP | XP(*) | XP(*)+MDP | XP(*)+ER |
|---|---|---|---|---|---|
| IPPO | $\mathbf{24.04 \pm 0.02}$ | $4.02 \pm 0.17$ | $0.12 \pm 0.03$ | $0.14 \pm 0.03$ | $0.10 \pm 0.03$ |
| IPPO + $OP^{\Phi^{MDP}}$ | $23.81 \pm 0.03$ | $8.61 \pm 0.17$ | $8.14 \pm 0.15$ | $8.70 \pm 0.16$ | $9.91 \pm 0.14$ |
| IPPO + $OP^{\Phi^{ER}}$ | $23.74 \pm 0.03$ | $\mathbf{21.64 \pm 0.07}$ | $\mathbf{22.03 \pm 0.05}$ | $\mathbf{22.50 \pm 0.06}$ | $\mathbf{22.25 \pm 0.05}$ |

## 5 RELATED WORK

Extensive research exists on coordination in multi-agent systems, particularly in zero-shot coordination. Methods like Hu et al. (2020); Muglich et al. (2022a) use Dec-POMDP symmetries to avoid incompatible policies, while Hu et al. (2021) rely on environment dynamics for grounded policies. Diversity-based approaches also leverage known symmetries and simulator access (Cui et al., 2023; Lupu et al., 2021). In contrast, ER symmetries can be learned from agent-environment interactions without privileged information, enabling grounded signaling and effective coordination in concurrent environments (see Sections 4.2 and 4.3).

In single-agent settings, symmetry has been shown to reduce sample complexity in RL (Van der Pol et al., 2020; Zhu et al., 2022; Nguyen et al., 2024). In multi-agent systems, symmetries reduce policy space complexity and help agents identify equivalent strategies (van der Pol et al., 2021; Muglich et al., 2022a). However, many methods require explicit knowledge of symmetries, or rely on predefined groups (Abreu et al., 2023; Yu et al., 2024; Nguyen et al., 2024). Our work generalizes these approaches by introducing ER symmetries, which do not require prior symmetry knowledge or equivariant networks, and can be learned directly through environment interactions.

Our work relates to value-based abstraction, which groups states or observations with similar value functions. Rezaei-Shoshtari et al. (2022) use lax-bisimulation to learn MDP homomorphisms, while Grimm et al. (2021) learn a model of the underlying MDP for value-based planning. In contrast, we focus on symmetries in the policy space that preserve expected return. ER symmetries are conceptually related to $Q^*$-irrelevance abstractions (Li et al., 2006) in that both aim to preserve the optimal value function of an MDP. However, whereas $Q^*$-irrelevance abstractions reduce complexity by aggregating states, ER symmetries form a group that acts bijectively on the policy space, transforming optimal policies into other policies with the same expected return.

## 6 CONCLUSION

This paper defined expected return symmetries—a group whose action preserves policy expected return. We demonstrated that the symmetries in this group can be learned purely from interactions with the environment and without requiring privileged environment information. We demonstrated that this symmetry class significantly enhances zero-shot coordination, significantly outperforming traditional Dec-POMDP symmetries, which are a subset of this group. Importantly, we showed that expected return symmetries are effective in challenging settings where state-of-the-art ZSC methods, such as off-belief learning (Hu et al., 2021) in Hanabi or approaches based on cognitive hierarchies, either fail completely (e.g., in the lever game and cat/dog environments) or face difficulties in their application (e.g., Overcooked V2).

One major limitation of our approach is that we constrain the search for symmetries to bijections over the action and observation spaces. While this works well in many settings, as shown in our experiments, there are environments, e.g. the two-lever game, in which this limited expressivity cannot provide enough diversity within the equivalence classes of policies that are optimal w.r.t. OP with $\Phi^{ERS}$, to prevent coordination failure in ZSC. Another limitation is that the success of our method is heavily dependent on the type of policies which are used to learn the ER symmetries.

Our work opens several avenues for future research. One direction is to explore the use of expected return symmetries in ad-hoc teamwork or single-agent settings. Another is to investigate broader classes of symmetries, beyond the ones that arise from bijections on the actions and observations.

ACKNOWLEDGEMENTS

Darius Muglich is supported by Toshiba Research, as well as the EPSRC centre for Doctoral Training in Autonomous and Intelligent Machines and Systems (EP/S024050/1). Johannes Forkel is funded by the UKI grant EP/Y028481/1 (originally selected for funding by the ERC). Jakob Foerster is partially funded by the UKI grant EP/Y028481/1. Jakob Foerster is also supported by the JPMC Research Award and the Amazon Research Award. EP/Y028481/1

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

## A EXPERIMENTAL SETUP

For expected return symmetry discovery in the three-lever game, each ERS agent trains 20 self-play optimal policies using IQL over 10000 episodes, with an $\epsilon = 0.1$ $\epsilon$-greedy behaviour policy and a learning rate of 0.1. The computed $Q$-values are then used to construct Boltzmann exploratory policies, with temperature $\alpha = 1$. Then for each pair of local action permutations, the ERS agent maximizes the objective 9, by searching over each pair of local observation permutations. The best 6 quadruples of two local action permuations and two local observation permuations are kept as learned expected return symmetries. Each of 5 ERS agents exactly finds the 6 Dec-POMDP symmetries.

For expected return symmetry discovery in cat/dog, we use a temperature of $\alpha = 1/2.667$ for Boltzmann exploration to promote sufficient exploration of different actions. We use a constant baseline function with value 9.5 for the policy gradient. We use a learning rate of 0.01 and 2000 episodes for each inner loop of Algorithm 1.

For expected return symmetry discovery in Overcooked V2 and Hanabi, we parameterize $\hat{\phi}_{\mathcal{O},\theta}$ as a two hidden layer, feedforward neural network, with each linear layer intialized as a $|\mathcal{O}^i|$-dimensional identity matrix; this choice of initialization is necessary as the symmetry discovery is highly initialization sensitive. We apply ReLU to the final output of the network to promote sparsity in the representation. We build on top of the environment implementation and baseline algorithms in JaxMARL (Rutherford et al., 2023).

For Hanabi, we run each inner loop of Algorithm 2 for $2.25e9$ timesteps across vectorized Hanabi environments. As per Equation 10 we use $\lambda_1 = 0.65$, $\lambda_2 = 2.5e-9$. $\phi_{\mathcal{A},\theta}$ is a fixed action transposition for each learned symmetry. We use a temperature of $\alpha = 1$ for Boltzmann exploration.

For Overcooked V2, we run each inner loop of Algorithm 3 for $1.5e8$ timesteps. $\phi_{\mathcal{A},\theta}$ is a learned affine map. We use a temperature of $\alpha = 1.1$ for Boltzmann exploration.

For both Hanabi and Overcooked V2, we use PPO and Generalized Advantage Estimation. For Hanabi, we use 4 epochs, 1024 environments per pretrained policy, 128 environment steps per update, 4 minibatches, $\gamma = 0.99$, GAE Lambda $= 0.95$, CLIP EPS $= 0.2$, VF COEFF = 0.5, MAX GRAD NORM $= 0.5$, a learning rate of $1e-5$ and a linear learning rate annealing schedule. For Overcooked V2, we use 4 epochs, 256 environments, 256 environment steps per update, 64 minibatches, $\gamma = 0.99$, GAE Lambda $= 0.95$, CLIP EPS $= 0.2$, VF COEFF $= 0.5$, MAX GRAD NORM $= 0.25$, a learning rate of $1e-5$ with no annealing.

Methods for Hanabi and Overcooked V2 were ran on A40 and L40 GPUs.

Code for Cat/Dog:

```
https://colab.research.google.com/drive/1enEW6cjnzTbM9sTtHlD2-vowYh9NDlhc?
usp=sharing
```

Code for Iterated Three-Lever Game:

```
https://colab.research.google.com/drive/1T9LpOkLDBl9BBkjzUelXKND6U8dOvfXV
```

Code for Hanabi/Overcooked V2:

```
https://github.com/gfppoy/expected-return-symmetries/tree/main
```

# B PROOFS

## B.1 GROUP PROPERTIES

**Theorem.** $\Phi^{\text{ER}} := \left\{ \phi \in \Psi \;\middle|\; \phi\big(\Pi_*^\alpha\big) = \Pi_*^\alpha \right\}$ forms a group under function composition.

*Proof.* To show that $\Phi^{\text{ER}}$ forms a group under function composition, we verify the group axioms: closure, associativity, identity, and inverses.

Closure: For any $\phi_1, \phi_2 \in \Phi^{\text{ER}}$, we need to show that $\phi_1 \circ \phi_2 \in \Phi^{\text{ER}}$. For the composition $\phi_1 \circ \phi_2$, we see that

$$(\phi_1 \circ \phi_2)(\Pi_*^\alpha) = \phi_1(\phi_2(\Pi_*^\alpha)) = \phi_1(\Pi_*^\alpha) = \Pi_*^\alpha.$$

which shows that $\phi_1 \circ \phi_2 \in \Phi^{\text{ER}}$.

Associativity: Function composition is associative, so for any $\phi_1, \phi_2, \phi_3 \in \Phi^{\text{ER}}$:

$$(\phi_1 \circ \phi_2) \circ \phi_3 = \phi_1 \circ (\phi_2 \circ \phi_3).$$

Thus, associativity holds.

Identity: The identity function $\mathbf{Id} \in \Psi$ satisfies $\mathbf{Id}(\pi) = \pi$ for all $\pi \in \Pi$, and thus $\mathbf{Id} \in \Phi^{\text{ER}}$ and acts as the identity element.

Inverses: We let $\phi \in \Phi^{\text{ER}}$. Since $\Psi$ is a finite group, there exists a positive integer $k$ such that $\phi^k = \mathbf{Id}$, and thus $\phi^{-1} = \phi^{k-1}$. Since $\phi^{k-1} \in \Phi^{\text{ER}}$ due to closure, we see that $\phi^{-1} \in \Phi^{\text{ER}}$.

Since $\Phi^{\text{ER}}$ satisfies closure, associativity, identity, and inverses, it forms a group under function composition. $\qquad\square$

**Theorem** (Dec-POMDP Symmetry Expected Return Invariance)**.** For any Dec-POMDP symmetry $\phi \in \Phi^{\text{MDP}}$, and any joint policy $\pi$, it holds that

$$J(\pi) = J(\phi(\pi)).$$

*Proof.* We prove by induction on the time step $t = 0, 1, \ldots, H$ that for every action–observation history (AOH) $\tau_t$,

$$V_\pi(\tau_t) = V_{\phi(\pi)}(\phi(\tau_t)).$$

Since the initial AOH is the empty history (i.e., $\tau_0 = \phi(\tau_0)$), this immediately implies

$$J(\pi) = V_\pi(\tau_0) = V_{\phi(\pi)}(\phi(\tau_0)) = J(\phi(\pi)).$$

**Base Case ($t = H$):** At time $H$ the game terminates, so by definition,

$$V_\pi(\tau_H) = 0 \quad \text{and} \quad V_{\phi(\pi)}(\phi(\tau_H)) = 0.$$

**Inductive Hypothesis:** Assume that for some $t$ with $t < H$, the equality

$$V_\pi(\tau_{t+1}) = V_{\phi(\pi)}(\phi(\tau_{t+1}))$$

holds for every AOH $\tau_{t+1}$.

**Inductive Step ($t$):** Consider an arbitrary AOH $\tau_t$. By the Bellman equation, the value function for $\pi$ is

$$V_\pi(\tau_t) = \sum_{s_t \in \mathcal{S}} \mathcal{B}_\pi(s_t \mid \tau_t) \sum_{a_t \in \mathcal{A}} \pi(a_t \mid \tau_t) \sum_{s_{t+1} \in \mathcal{S}} \mathcal{T}(s_{t+1} \mid s_t, a_t)$$

$$\times \left[ \mathcal{R}(s_{t+1}, a_t) + \gamma \sum_{o_{t+1} \in \mathcal{O}} \mathcal{U}(o_{t+1} \mid s_{t+1}, a_t) V_\pi(\tau_{t+1}) \right].$$

Similarly, the value function for the transformed policy $\phi(\pi)$ on the transformed history $\phi(\tau_t)$ is

$$
\begin{aligned}
V_{\phi(\pi)}(\phi(\tau_t)) = \sum_{s_t \in \mathcal{S}} \mathcal{B}_{\phi(\pi)}(\phi(s_t) \mid \phi(\tau_t)) \sum_{a_t \in \mathcal{A}} \phi(\pi)(\phi(a_t) \mid \phi(\tau_t)) \\
\times \sum_{s_{t+1} \in \mathcal{S}} \mathcal{T}(\phi(s_{t+1}) \mid \phi(s_t), \phi(a_t)) \\
\times \left[ \mathcal{R}(\phi(s_{t+1}), \phi(a_t)) + \gamma \sum_{o_{t+1} \in \mathcal{O}} \mathcal{U}(\phi(o_{t+1}) \mid \phi(s_{t+1}), \phi(a_t)) V_{\phi(\pi)}(\phi(\tau_{t+1})) \right].
\end{aligned}
$$

By the symmetry properties (e.g., Equations 2 and 3), we have

$$
\mathcal{B}_{\phi(\pi)}(\phi(s_t) \mid \phi(\tau_t)) = \mathcal{B}_\pi(s_t \mid \tau_t), \quad \phi(\pi)(\phi(a_t) \mid \phi(\tau_t)) = \pi(a_t \mid \tau_t),
$$

and the invariance of $\mathcal{T}, \mathcal{R},$ and $\mathcal{U}$ under $\phi$. Thus,

$$
\begin{aligned}
V_{\phi(\pi)}(\phi(\tau_t)) = \sum_{s_t \in \mathcal{S}} \mathcal{B}_\pi(s_t \mid \tau_t) \sum_{a_t \in \mathcal{A}} \pi(a_t \mid \tau_t) \sum_{s_{t+1} \in \mathcal{S}} \mathcal{T}(s_{t+1} \mid s_t, a_t) \\
\times \left[ \mathcal{R}(s_{t+1}, a_t) + \gamma \sum_{o_{t+1} \in \mathcal{O}} \mathcal{U}(o_{t+1} \mid s_{t+1}, a_t) V_{\phi(\pi)}(\phi(\tau_{t+1})) \right].
\end{aligned}
$$

By the inductive hypothesis $V_{\phi(\pi)}(\phi(\tau_{t+1})) = V_\pi(\tau_{t+1})$, so it follows that

$$
V_{\phi(\pi)}(\phi(\tau_t)) = V_\pi(\tau_t).
$$

This completes the inductive step.

By mathematical induction, the equality $V_\pi(\tau_t) = V_{\phi(\pi)}(\phi(\tau_t))$ holds for all $t = 0, 1, \ldots, H$, and hence

$$
J(\pi) = V_\pi(\tau_0) = V_{\phi(\pi)}(\phi(\tau_0)) = J(\phi(\pi)).
$$

$\square$

**Theorem** (Dec-POMDP Symmetry Boltzmann Exploratory Invariance). Let $\phi \in \Phi^{\mathrm{MDP}}$ be a Dec-POMDP symmetry and let $\pi \in \Pi^\alpha$ be a Boltzmann exploratory joint policy with temperature $\alpha > 0$. Then, the transformed policy $\phi(\pi)$ is also Boltzmann exploratory, i.e.,

$$
\phi(\pi)^i\left(\phi(a_t^i) \mid \phi(\tau_t^i)\right) = \frac{\exp\left(\frac{Q_{\phi(\pi)}^i\left(\phi(\tau_t^i), \phi(a_t^i)\right)}{\alpha}\right)}{\sum_{a^i \in \phi^{-1}\left(\mathcal{A}^i(\phi(\tau_t^i))\right)} \exp\left(\frac{Q_{\phi(\pi)}^i\left(\phi(\tau_t^i), \phi(a^i)\right)}{\alpha}\right)}.
$$

Moreover, it holds that

$$
Q_{\phi(\pi)}^i\left(\phi(\tau_t^i), \phi(a_t^i)\right) = Q_\pi^i(\tau_t^i, a_t^i),
$$

for all agents $i$, times $t$, AOHs $\tau_t^i$, and local actions $a_t^i$.

*Proof.* We first prove by induction over the time step $t$ that for every agent $i$,

$$
Q_{\phi(\pi)}^i\left(\phi(\tau_t^i), \phi(a_t^i)\right) = Q_\pi^i(\tau_t^i, a_t^i).
$$

**Base Case ($t = H$):**
At terminal time $H$, the episode ends. Hence, by definition we have

$$
Q_\pi^i(\tau_H^i, a_H^i) = 0 \quad \text{and} \quad Q_{\phi(\pi)}^i\left(\phi(\tau_H^i), \phi(a_H^i)\right) = 0.
$$

Thus, the base case holds.

**Inductive Hypothesis:**
Assume that for some $t < H$, and for all agents $i$, all local histories $\tau_{t+1}^i$, and actions $a_{t+1}^i$, the following holds:

$$
Q_{\phi(\pi)}^i\left(\phi(\tau_{t+1}^i), \phi(a_{t+1}^i)\right) = Q_\pi^i(\tau_{t+1}^i, a_{t+1}^i).
$$

**Inductive Step ($t$):**
For a given local AOH $\tau_t^i$ and local action $a_t^i$, the Bellman equation gives

$$
\begin{aligned}
&Q_\pi^i(\tau_t^i, a_t^i) \\
&= \sum_{s_t, \tau_t} \mathcal{B}_\pi(s_t, \tau_t \mid \tau_t^i) \sum_{a_t^{-i}} \pi^{-i}(a_t^{-i} \mid \tau_t^{-i}) \sum_{s_{t+1}} \mathcal{T}(s_{t+1} \mid s_t, a_t) \\
&\quad \times \left[ \mathcal{R}(s_{t+1}, a_t) + \gamma \sum_{o_{t+1}^i} \mathcal{U}^i(o_{t+1}^i \mid s_{t+1}, a_t) \sum_{a_{t+1}^i \in \mathcal{A}^i(\tau_{t+1}^i)} \pi^i(a_{t+1}^i \mid \tau_{t+1}^i) Q_\pi^i(\tau_{t+1}^i, a_{t+1}^i) \right].
\end{aligned}
\tag{12}
$$

Here $\pi^{-i}$, $a_t^{-i}$, $\tau_t^{-i}$, denote the parts of the joint policy $\pi$, joint action $a_t$, and joint AOH $\tau_t$, which do not belong to agent $i$.

Similarly, under the transformed policy $\phi(\pi)$, the $Q$-function for agent $i$ is given by

$$
\begin{aligned}
&Q_{\phi(\pi)}^i\big(\phi(\tau_t^i), \phi(a_t^i)\big) \\
&= \sum_{s_t, \tau_t \in \mathcal{S}} \mathcal{B}_{\phi(\pi)}\big(\phi(s_t), \phi(\tau_t) \mid \phi(\tau_t^i)\big) \sum_{a_t^{-i}} \phi(\pi)^{-i}\big(\phi(a_t^{-i}) \mid \phi(\tau_t^{-i})\big) \sum_{s_{t+1}} \mathcal{T}\big(\phi(s_{t+1}) \mid \phi(s_t), \phi(a_t)\big) \\
&\quad \times \left[ \mathcal{R}\big(\phi(s_{t+1}), \phi(a_t)\big) + \gamma \sum_{o_{t+1}^i} \mathcal{U}^i\big(\phi(o_{t+1}^i) \mid \phi(s_{t+1}), \phi(a_t)\big) \right. \\
&\quad \left. \times \sum_{a_{t+1}^i \in \phi^{-1}\big(\mathcal{A}^i(\phi(\tau_{t+1}^i))\big)} \phi(\pi)^i\big(\phi(a_{t+1}^i) \mid \phi(\tau_{t+1}^i)\big) Q_{\phi(\pi)}^i\big(\phi(\tau_{t+1}^i), \phi(a_{t+1}^i)\big) \right].
\end{aligned}
\tag{13}
$$

Since $\phi$ is a Dec-POMDP symmetry, Equation 3 states that

$$
\begin{aligned}
\mathcal{T}\big(\phi(s_{t+1}) \mid \phi(s_t), \phi(a_t)\big) &= \mathcal{T}(s_{t+1} \mid s_t, a_t), \\
\mathcal{U}\big(\phi(o_{t+1}) \mid \phi(s_{t+1}), \phi(a_t)\big) &= \mathcal{U}(o_{t+1} \mid s_{t+1}, a_t), \\
\mathcal{R}\big(\phi(s_{t+1}), \phi(a_t)\big) &= \mathcal{R}(s_{t+1}, a_t).
\end{aligned}
$$

Hence, for the belief $\mathcal{B}_\pi$ it holds that

$$
\mathcal{B}_{\phi(\pi)}\big(\phi(s_t), \phi(\tau_t) \mid \phi(\tau_t^i)\big) = \mathcal{B}_\pi(s_t, \tau_t \mid \tau_t^i).
$$

By definition of $\phi(\pi)$ we see that

$$
\phi(\pi)^{-i}\big(\phi(a_t^{-i}) \mid \phi(\tau_t^{-i})\big) = \pi^{-i}(a_t^{-i} \mid \tau_t^{-i}).
$$

Moreover, by the inductive hypothesis,

$$
Q_{\phi(\pi)}^i\big(\phi(\tau_{t+1}^i), \phi(a_{t+1}^i)\big) = Q_\pi^i(\tau_{t+1}^i, a_{t+1}^i).
$$

Finally, it holds that

$$
\phi^{-1}\big(\mathcal{A}^i(\phi(\tau_t^i))\big) = \mathcal{A}^i(\tau_t^i),
$$

as this is a necessary condition on $\phi$, such that $\phi(\pi)$ is well-defined for all $\pi$.

Substituting all of the above equations into Equation 13, we obtain:

$$
Q_{\phi(\pi)}^i\big(\phi(\tau_t^i), \phi(a_t^i)\big) = Q_\pi^i(\tau_t^i, a_t^i).
$$

This completes the induction step.

Therefore, for any agent $i$, local history $\tau_t^i$, and action $a_t^i$, the transformed policy satisfies

$$\phi(\pi)^i\Big(\phi(a_t^i)\,\Big|\,\phi(\tau_t^i)\Big) = \pi^i(a_t^i|\tau_t^i) = \frac{\exp\left(\frac{Q_\pi^i(\tau_t^i,a_t^i)}{\alpha}\right)}{\sum_{a^i\in\mathcal{A}^i(\tau_t^i)}\exp\left(\frac{Q_\pi^i(\tau_t^i,a^i)}{\alpha}\right)}$$

$$= \frac{\exp\left(\frac{Q_{\phi(\pi)}^i\big(\phi(\tau_t^i),\phi(a_t^i)\big)}{\alpha}\right)}{\sum_{a^i\in\phi^{-1}\big(\mathcal{A}^i(\phi(\tau_t^i))\big)}\exp\left(\frac{Q_{\phi(\pi)}^i\big(\phi(\tau_t^i),\phi(a^i)\big)}{\alpha}\right)}.$$

which shows that $\phi(\pi)\in\Pi^\alpha$.

$\square$

## B.2 Policy-Gradient Based Boltzmann-Exploratory Policies

Recall that for a decentralized partially observable setting, each agent $i$ acts based on its local action–observation history (AOH) $\tau_t^i$. The value of $\tau_t^i$, when assuming that all agents follow their respective local policies in $\pi$, is given by

$$V_\pi^i(\tau_t^i) = \sum_{a^i\in\mathcal{A}^i(\tau_t^i)} \pi^i(a_t^i\mid\tau_t^i)\,Q_\pi^i(\tau_t^i,a_t^i),$$

where $Q_\pi^i(\tau_t^i,a_t^i)$ is the local action–value function for agent $i$, when assuming that all agents follow their respective local policies in $\pi$.

In practice, actor–critic methods update the critic and actor on different timescales. With appropriate learning rate schedules, the local value estimates $Q_\pi^i(\tau_t^i,a_t^i)$ converge and approximately satisfy the Bellman equations. In this regime, we may treat $Q_\pi^i(\tau_t^i,a_t^i)$ as fixed when optimizing the policy. Then, for agent $i$, the entropy-regularized objective becomes

$$J_{\tau_t^i}(\pi^i) = \sum_{a^i\in\mathcal{A}^i(\tau_t^i)} \pi^i(a_t^i\mid\tau_t^i)\,Q_\pi^i(\tau_t^i,a_t^i) + \alpha\,H\big(\pi^i(\cdot\mid\tau_t^i)\big),$$

with the entropy given by

$$H\big(\pi^i(\cdot\mid\tau_t^i)\big) = -\sum_{a^i\in\mathcal{A}^i(\tau_t^i)} \pi^i(a_t^i\mid\tau_t^i)\log\pi^i(a_t^i\mid\tau_t^i).$$

Because $Q_\pi^i(\tau_t^i,a_t^i)$ is treated as fixed and the entropy function is strictly concave, the local objective $J_{\tau_t^i}(\pi^i)$ is strictly concave in $\pi^i(\cdot\mid\tau_t^i)$. Hence, its unique maximizer can be computed independently for each agent and each local history by solving a Lagrangian optimization problem.

The following theorem formalizes that the unique maximizer of $J_{\tau_t^i}(\pi^i)$ is the Boltzmann (softmax) policy.

**Theorem** (Local Boltzmann Policy from Entropy-Regularized Objective). For a fixed local AOH $\tau_t^i$ of agent $i$, the optimal local policy that maximizes

$$J_{\tau_t^i}(\pi^i) = \sum_{a^i\in\mathcal{A}^i(\tau_t^i)} \pi^i(a_t^i\mid\tau_t^i)\,Q_\pi^i(\tau_t^i,a_t^i) + \alpha\,H\big(\pi^i(\cdot\mid\tau_t^i)\big)$$

subject to

$$\sum_{a^i\in\mathcal{A}^i(\tau_t^i)} \pi^i(a_t^i\mid\tau_t^i) = 1,$$

is given by

$$\pi^i(a_t^i\mid\tau_t^i) = \frac{\exp\left(\frac{Q_\pi^i(\tau_t^i,a_t^i)}{\alpha}\right)}{\sum_{a'\in\mathcal{A}^i(\tau_t^i)}\exp\left(\frac{Q_\pi^i(\tau_t^i,a')}{\alpha}\right)}.$$

*Proof.* Assume that for the fixed local AOH $\tau_t^i$, the value estimates $Q_\pi^i(\tau_t^i, a_t^i)$ are treated as constant. Define the Lagrangian

$$\mathcal{L}(\pi^i, \lambda) = \sum_{a^i \in \mathcal{A}^i(\tau_t^i)} \pi^i(a_t^i \mid \tau_t^i) \, Q_\pi^i(\tau_t^i, a_t^i)$$

$$- \alpha \sum_{a^i \in \mathcal{A}^i(\tau_t^i)} \pi^i(a_t^i \mid \tau_t^i) \log \pi^i(a_t^i \mid \tau_t^i) + \lambda \left( \sum_{a^i \in \mathcal{A}^i(\tau_t^i)} \pi^i(a_t^i \mid \tau_t^i) - 1 \right).$$

Taking the derivative with respect to $\pi^i(a_t^i \mid \tau_t^i)$ for each $a^i \in \mathcal{A}^i(\tau_t^i)$ yields

$$\frac{\partial \mathcal{L}}{\partial \pi^i(a_t^i \mid \tau_t^i)} = Q_\pi^i(\tau_t^i, a_t^i) - \alpha\big(\log \pi^i(a_t^i \mid \tau_t^i) + 1\big) + \lambda = 0.$$

Rearrange this equation to obtain

$$\log \pi^i(a_t^i \mid \tau_t^i) = \frac{Q_\pi^i(\tau_t^i, a_t^i) + \lambda - \alpha}{\alpha}.$$

Exponentiating both sides gives

$$\pi^i(a_t^i \mid \tau_t^i) = \exp\left( \frac{Q_\pi^i(\tau_t^i, a_t^i)}{\alpha} \right) \exp\left( \frac{\lambda - \alpha}{\alpha} \right).$$

Since the term $\exp\left( \frac{\lambda - \alpha}{\alpha} \right)$ is independent of $a^i$, it is determined by the normalization constraint:

$$\sum_{a^i \in \mathcal{A}^i(\tau_t^i)} \pi^i(a_t^i \mid \tau_t^i) = \exp\left( \frac{\lambda - \alpha}{\alpha} \right) \sum_{a^i \in \mathcal{A}^i(\tau_t^i)} \exp\left( \frac{Q_\pi^i(\tau_t^i, a_t^i)}{\alpha} \right) = 1.$$

Defining

$$Z^i(\tau_t^i) = \sum_{a^i \in \mathcal{A}^i(\tau_t^i)} \exp\left( \frac{Q_\pi^i(\tau_t^i, a_t^i)}{\alpha} \right),$$

we have

$$\exp\left( \frac{\lambda - \alpha}{\alpha} \right) = \frac{1}{Z^i(\tau_t^i)}.$$

Thus, the unique maximizer is given by

$$\pi^i(a_t^i \mid \tau_t^i) = \frac{\exp\left( \frac{Q_\pi^i(\tau_t^i, a_t^i)}{\alpha} \right)}{Z^i(\tau_t^i)} = \frac{\exp\left( \frac{Q_\pi^i(\tau_t^i, a_t^i)}{\alpha} \right)}{\sum_{a' \in \mathcal{A}^i(\tau_t^i)} \exp\left( \frac{Q_\pi^i(\tau_t^i, a')}{\alpha} \right)}.$$

$\square$

Note the $Q$-functions above can be provided by a critic network or be estimated via empirical returns. When the local $Q_\pi^i(\tau_t^i, a_t^i)$ estimates have converged, the local objective decouples over $\tau_t^i$ and its unique maximizer is the Boltzmann (softmax) policy. This motivates our definition of the set of Boltzmann exploratory policies,

$$\Pi^\alpha := \left\{ \pi \in \Pi \; \middle| \; \pi^i(a_t^i \mid \tau_t^i) = \frac{\exp\left( \frac{Q_\pi^i(\tau_t^i, a_t^i)}{\alpha} \right)}{\sum_{a^i \in \mathcal{A}^i(\tau_t^i)} \exp\left( \frac{Q_\pi^i(\tau_t^i, a^i)}{\alpha} \right)} \; , \; \forall\, a_t^i, \, \forall\, \tau_t^i, \, \forall\, i = 1, \dots, n \right\},$$

and, correspondingly, our definition of the set of self-play optimal Boltzmann-exploratory policies

$$\Pi_*^\alpha := \arg\max_{\pi \in \Pi^\alpha} J(\pi).$$

Finally, we define expected return symmetries,

$$\Phi^{\mathrm{ER}} := \left\{ \phi \in \Psi \; \middle| \; \forall\, \pi \in \Pi_*^\alpha : \phi(\pi) \in \Pi_*^\alpha \right\},$$

as the subset of transformations preserving self-play optimality of Boltzmann-exploratory policies.

## C  Why Boltzmann-Exploratory Policies are Necessary

Defining expected return symmetries in terms of Boltzmann-exploratory policies, rather than just $\epsilon$-soft policies, is necessary. If defined via $\epsilon$-soft policies, permuting any set of actions which are suboptimal in self-play would be an expected return symmetry, since all suboptimal actions would be taken with the same probability. The following example illustrates this:

Consider the cat/dog game from Example 2. Boltzmann exploration is necessary to disambiguate for Alice the actions of removing the barrier and bailing: simple $\epsilon-$greedy exploration does not suffice, since we would then still use both these suboptimal actions with equal probability, and so the expected return remains the same under the relabeling of these actions. Thus, under $\epsilon-$greedy exploration, permuting the labels of the actions for removing the barrier and bailing is an expected return symmetry. This, however, is undesirable because we want $\mathrm{OP}^{\Phi^{\mathrm{ER}}}$-optimal policies to be able to disambiguate these very different actions. Under Boltzmann exploration, Alice takes the reveal action more often than the bail action, and so relabeling these actions lowers expected return and is thus not an expected return symmetry.

## D  Learned Transformations Satisfy Group Properties

This section analyzes the learned ER symmetries for their group properties.

We first train six (near-)optimal IPPO policies with independent seeds as $\Pi'$, obtaining a mean expected return of $\mathbb{E}_{\pi \sim \Pi'}[J(\pi)] = 24.04 \pm 0.02$. Next, we randomly select 64 local-action-space transpositions, and learn the corresponding $\{\phi_{\mathcal{O},\theta_l}\}_{l=1}^{64}$ via optimizing Equation 9 / Algorithm 1. That is, we fix the same action transposition for each local policy, significantly constraining the search. This is a significant undersampling of the 190 possible transpositions on each local action space, yet as we show below we still learn effective ER symmetries for coordination. We then save the 11 best transformations (those that maximize Equation 9) as unregularized ER symmetries. These are used to maximize Equation 10 / Algorithm 2 on another set of 64 random transpositions, now enforcing compositional closure and invertibility. The 11 best are saved as regularized ER symmetries.

Table 2: Comparison of 11 unregularized and 11 regularized ER symmetries applied to 6 unseen optimal policies ($|\Pi_{\mathrm{unseen}}| = 6$). Regularization enforces compositionality and invertibility. For $k = 1, 2, 3$, let $J_k = \mathbb{E}_{\phi_i \sim \Phi} \mathbb{E}_{\pi \sim \Pi_{\mathrm{unseen}}}[J((\phi_1 \circ \cdots \circ \phi_k)(\pi))]$ denote the expected return after composing $k$ randomly sampled transformations. We report Single Transform. ($J_1$), Double Comp. ($J_2$), and Triple Comp. ($J_3$). Relative Reconstruction Loss measures approximate invertibility (lower is better): $\mathbb{E}_{\pi \sim \Pi_{\mathrm{unseen}}} \mathbb{E}_{\tau \sim \pi}[\frac{||\tau - \phi_{\mathcal{O}}^2(\tau)||}{||\tau||}]$, using the $\ell_2$ norm for AOH vectors. Recall $\mathbb{E}_{\pi \sim \Pi'}[J(\pi)] = 24.04 \pm 0.02$.

|  | Single Transform. | Double Comp. | Triple Comp. | Rel. Rec. Loss |
|---|---|---|---|---|
| **Unreg.** | $22.88 \pm 0.07$ | $21.10 \pm 0.09$ | $20.36 \pm 0.11$ | $31.4\% \pm 1.8\%$ |
| **Reg.** | $23.32 \pm 0.05$ | $22.16 \pm 0.07$ | $20.94 \pm 0.12$ | $16.7\% \pm 0.18\%$ |

Table 2 shows that up to minor deviations in expected return preservation, the learned ER symmetries still approximately satisfy closure under composition. In addition, when invertibility is enforced, the relative reconstruction loss decreases substantially (a lower relative reconstruction loss tells us applying the transformation twice brings us closer to the original AOH, suggesting approximate invertibility). We conclude that the learned ER symmetries, especially the regularized ones, approximately satisfy the desired group-theoretic properties.

# E  ALGORITHMS

---

**Algorithm 1** Learning Expected Return Symmetries with Policy Gradients (without enforcing compositionality nor invertibility) . . . Optimization of Equation 9

---

1: **Input:** A Dec-POMDP, a set $\Pi'$ of joint policies in it, a parameterization $\phi_{\mathcal{O},\theta}, \theta \in \Theta$, a learning rate $\eta > 0$, $l$ for the number of top transformations to save
2: Initialize list of top $l$ average expected returns: $\bar{J}_{\text{top}} = [-\infty, \ldots, -\infty]$ (length $l$)
3: Initialize list of top $l$ transformations: $\phi_{\text{top}} = [\emptyset, \ldots, \emptyset]$ (length $l$)
4: **for** each tuple of local action transpositions $\phi_{\mathcal{A}} \in \prod_{i=1}^{n} \text{Transpositions}(\mathcal{A}^i)$ **do**
5:     Initialize $\phi_{\mathcal{O},\theta}$ with random parameters $\theta \in \Theta$
6:     **while** not converged **do**
7:         **for** each policy $\pi \in \Pi'$ **do**
8:             Sample a batch $\mathcal{B}$ of joint AOHs, and the corresponding sequences of returns, using the transformed policy $\phi_{\theta}(\pi) = (\phi_{\mathcal{A}}, \phi_{\mathcal{O},\theta})(\pi)$
9:             Compute advantage $A^{\phi_{\theta}(\pi)}(\tau_t, a_t)$ for all $t = 0, ..., H-1$, using any advantage function (e.g., TD, GAE)
10:             Compute policy gradient:

$$\nabla_{\theta} J(\phi_{\theta}(\pi)) \approx \frac{1}{|\mathcal{B}|} \sum_{\tau_H \in \mathcal{B}} \left[ \sum_{t=0}^{H-1} \nabla_{\theta} \log \phi_{\theta}(\pi)(a_t \mid \tau_t) A^{\phi_{\theta}(\pi)}(\tau_t, a_t) \right]$$

11:             Update parameters: $\theta \leftarrow \theta + \eta \nabla_{\theta} J(\phi_{\theta}(\pi))$
12:         **end for**
13:     **end while**
14:     Compute average expected return $\bar{J}_{\phi_{\theta}}$, where for every $\pi \in \Pi'$ the expected return $J(\phi_{\theta}(\pi))$ is approximated by the average return over a number of episodes:

$$\bar{J}_{\phi_{\theta}} \approx \frac{1}{|\Pi'|} \sum_{\pi \in \Pi'} J(\phi_{\theta}(\pi))$$

15:     Find the index of the lowest return in $\bar{J}_{\text{top}}$, say $i_{\min}$
16:     **if** $\bar{J}_{\phi_{\theta}} > \bar{J}_{\text{top}}[i_{\min}]$ **then**
17:         Replace the lowest return: $\bar{J}_{\text{top}}[i_{\min}] \leftarrow \bar{J}_{\phi_{\theta}}$
18:         Replace the corresponding transformation: $\phi_{\text{top}}[i_{\min}] \leftarrow (\phi_{\mathcal{O},\theta}, \phi_{\mathcal{A},\theta})$:
19:     **end if**
20: **end for**
21: **Output:** Set $\phi_{\text{top}}$ of the best $l$ learned transformations

---

---

**Algorithm 2** Learning Expected Return Symmetries (enforcing compositionality and invertibility) ... Optimization of Equation 10

---

1: **Input:** A Dec-POMDP, a set $\Pi'$ of joint policies in it, a parameterization $\phi_{\mathcal{O},\theta}, \theta \in \Theta$, a learning rate $\eta > 0$, transformations $\{\phi_1, \ldots, \phi_m\}$ obtained from Algorithm 1, $l$ for the number of top transformations to save, regularization weights $\lambda_1, \lambda_2$
2: Initialize list of top $l$ average expected returns: $\bar{J}_{\text{top}} = [-\infty, \ldots, -\infty]$ (length $l$)
3: Initialize list of top $l$ transformations: $\phi_{\text{top}} = [\emptyset, \ldots, \emptyset]$ (length $l$)
4: **for** each tuple of local action transpositions $\phi_{\mathcal{A}} \in \prod_{i=1}^{n} \text{Transpositions}(\mathcal{A}^i)$ **do**
5:     Initialize $\phi_{\mathcal{O},\theta}$ with random $\theta \in \Theta$
6:     **while** not converged **do**
7:         **for** each policy $\pi \in \Pi'$ **do**
8:             With probability $1 - \lambda_1$ set $\tilde{\phi}_\theta = \phi_\theta$, and with probability $\lambda_1$ sample $\phi_i, \phi_j \in \{\phi_1, \ldots, \phi_m\}$ and set $\tilde{\phi}_\theta = \phi_i \circ \phi_\theta \circ \phi_j$
9:             Sample a batch $\mathcal{B}$ of joint AOHs, and the corresponding sequences of returns, using the transformed policy $\tilde{\phi}_\theta(\pi)$
10:            Compute advantage $A^{\tilde{\phi}_\theta(\pi)}(\tau_t, a_t)$, for all $t = 0, ..., H-1$, using any advantage function (e.g., TD, GAE)
11:            Compute the invertibility regularization term $L(\theta) = \frac{1}{|\mathcal{O}|} \sum_{o \in \mathcal{O}} \left[ d(o, \phi_{\mathcal{O},\theta}^2(o))^2 \right]$, and its gradient $\nabla_\theta L(\theta)$
12:            Compute policy gradient:

$$\nabla_\theta J(\tilde{\phi}_\theta(\pi)) \approx \frac{1}{|\mathcal{B}|} \sum_{\tau_H \in \mathcal{B}} \left[ \sum_{t=0}^{H-1} \nabla_\theta \log \tilde{\phi}_\theta(\pi)(a_t \mid \tau_t) A^{\tilde{\phi}_\theta(\pi)}(\tau_t, a_t) \right]$$

13:            Update $\theta \leftarrow \theta + \eta \big( \nabla_\theta J(\tilde{\phi}_\theta(\pi)) - \lambda_2 \nabla_\theta L(\theta) \big)$
14:        **end for**
15:        Compute average expected return $\bar{J}_{\phi_\theta}$, where for every $\pi \in \Pi'$ the expected return $J(\phi_\theta(\pi))$ is approximated by the average return over a number of episodes

$$\bar{J}_{\phi_\theta} \approx \frac{1}{|\Pi'|} \sum_{\pi \in \Pi'} J(\phi_\theta(\pi))$$

16:    **end while**
17:    Find the index of the lowest return in $\bar{J}_{\text{top}}$, say $i_{\min}$
18:    **if** $\bar{J}_{\phi_\theta} > \bar{J}_{\text{top}}[i_{\min}]$ **then**
19:        Replace the lowest return: $\bar{J}_{\text{top}}[i_{\min}] \leftarrow \bar{J}_{\phi_\theta}$
20:        Replace the corresponding transformation: $\phi_{\text{top}}[i_{\min}] \leftarrow (\phi_{\mathcal{O},\theta}, \phi_{\mathcal{A},\theta})$
21:    **end if**
22: **end for**
23: **Output:** Set $\phi_{\text{top}}$ of the best $l$ learned transformations

---

---

**Algorithm 3** Learning Expected Return Symmetries through cross-play maximization between pairs of Policies ... Optimization of Equation 11

---

1: **Input:** A Dec-POMDP, a set $\Pi'$ of joint policies in it, a parameterization $\phi_{\mathcal{O},\theta}, \theta \in \Theta$, a learning rate $\eta > 0$, $l$ for the number of top transformations to save
2: Initialize list of top $l$ average expected returns: $\bar{J}_{\text{top}} = [-\infty, \dots, -\infty]$ (length $l$)
3: Initialize list of top $l$ transformations: $\phi_{\text{top}} = [\emptyset, \dots, \emptyset]$ (length $l$)
4: **for** each pair of joint policies $(\pi_i, \pi_j) \in \Pi' \times \Pi' \setminus \{(\pi, \pi) \mid \pi \in \Pi'\}$ **do**
5:     Initialize $\phi_{\mathcal{O},\theta}$ and $\phi_{\mathcal{A},\theta}$ with random parameters $\theta \in \Theta$
6:     **while** not converged **do**
7:         Sample a batch $\mathcal{B}$ of joint AOHs, and the corresponding sequences of returns, using the transformed pair $(\pi_i^1, \phi_\theta(\pi_j^2))$
8:         Compute advantage $A^{(\pi_i^1, \phi_\theta(\pi_j^2))}(\tau_t, a_t)$, for all $t = 0, \dots, H-1$, using any advantage function (e.g., TD, GAE)
9:         Compute policy gradient:

$$\nabla_\theta J(\pi_i^1, \phi_\theta(\pi_j^2)) \approx \frac{1}{|\mathcal{B}|} \sum_{\tau_H \in \mathcal{B}} \left[ \sum_{t=0}^{H-1} \nabla_\theta \log \phi_\theta(\pi_j^2)(a_t \mid \tau_t) A^{(\pi_i^1, \phi_\theta(\pi_j^2))}(\tau_t, a_t) \right]$$

10:         Update parameters: $\theta \leftarrow \theta + \eta \nabla_\theta J(\pi_i^1, \phi_\theta(\pi_j^2))$
11:         Compute average return $\bar{J}_{(\pi_i^1, \phi_\theta(\pi_j^2))} \approx J(\pi_i^1, \phi_\theta(\pi_j^2))$ over a number of episodes
12:     **end while**
13:     Find the index of the lowest return in $\bar{J}_{\text{top}}$, say $i_{\min}$
14:     **if** $\bar{J}_{(\pi_i^1, \phi_\theta(\pi_j^2))} > \bar{J}_{\text{top}}[i_{\min}]$ **then**
15:         Replace the lowest return: $\bar{J}_{\text{top}}[i_{\min}] \leftarrow \bar{J}_{(\pi_i^1, \phi_\theta(\pi_j^2))}$
16:         Replace the corresponding transformation: $\phi_{\text{top}}[i_{\min}] \leftarrow (\phi_{\mathcal{O},\theta}, \phi_{\mathcal{A},\theta})$
17:     **end if**
18: **end for**
19: **Output:** Set $\phi_{\text{top}}$ of the best $l$ learned transformations

---

## F  INTERPRETABILITY OF HANABI OP AGENTS

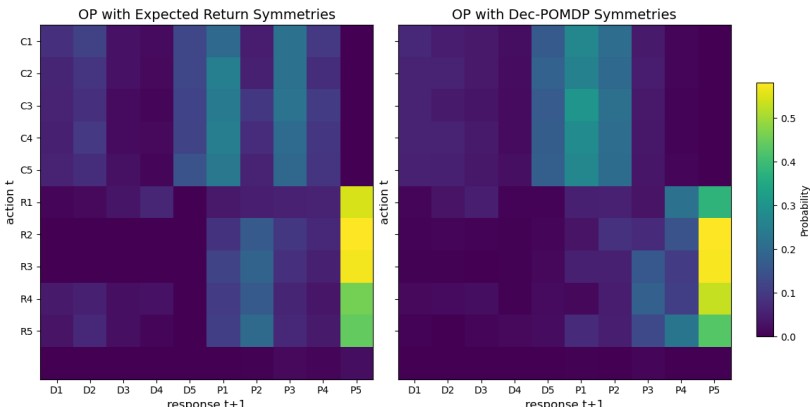

Figure 3: Conditional action matrices of $\text{OP}^{\Phi^{\text{MDP}}}$-optimal and $\text{OP}^{\Phi^{\text{ER}}}$-optimal policies; i.e., $P(a_t^i \mid a_{t-1}^j)$. We select the agent from both respective populations achieving the highest cross-play scores. We can see the $\text{OP}^{\Phi^{\text{ER}}}$-optimal policy more consistently uses a rank hint to signal playing the fifth card, whereas the $\text{OP}^{\Phi^{\text{MDP}}}$-optimal policy uses a similar convention but less consistently.

## G  HANABI

Hanabi is a cooperative card game that can be played with 2 to 5 people. Hanabi is a popular game, having been crowned the 2013 "Spiel des Jahres" award, a German industry award given to the best board game of the year. Hanabi has been proposed as an AI benchmark task to test models of cooperative play that act under partial information Bard et al. (2020). To date, Hanabi has one of the largest state spaces of all Dec-POMDP benchmarks.

The deck of cards in Hanabi is comprised of five colors (white, yellow, green, blue and red), and five ranks (1 through 5), where for each color there are three 1's, two each of 2's, 3's and 4's, and one 5, for a total deck size of fifty cards. Each player is dealt five cards (or four cards if there are 4 or 5 players). At the start, the players collectively have eight information tokens and three fuse tokens, the uses of which shall be explained presently.

In Hanabi, players can see all other players' hands but their own. The goal of the game is to play cards to collectively form five consecutively ordered stacks, one for each color, beginning with a card of rank 1 and ending with a card of rank 5. These stacks are referred to as fireworks, as playing the cards in order is meant to draw analogy to setting up a firework display.

We call the player whose turn it is the active agent. The active agent must conduct one of three actions:

- **Hint** - The active agent chooses another player to grant a hint to. A hint involves the active agent choosing a color or rank, and revealing to their chosen partner all cards in the partner's hand that satisfy the chosen color or rank. Performing a hint exhausts an information token. If the players have no information tokens, a hint may not be conducted and the active agent must either conduct a discard or a play.

- **Discard** - The active agent chooses one of the cards in their hand to discard. The identity of the discarded card is revealed to the active agent and becomes public information. Discarding a card replenishes an information token should the players have less than eight.

- **Play** - The active agent attempts to play one of the cards in their hand. The identity of the played card is revealed to the active agent and becomes public information. The active agent has played successfully if their played card is the next in the firework of its color to be played, and the played card is then added to the sequence. If a firework is completed, the players receive a new information token should they have less than eight. If the player

is unsuccessful, the card is discarded, without replenishment of an information token, and the players lose a fuse token.

The game ends when all three fuse tokens are spent, when the players successfully complete all five fireworks, or when the last card in the deck is drawn and all players take one last turn. If the game finishes by depletion of all fuse tokens (i.e. by "bombing out"), the players receive a score of 0. Otherwise, the score of the finished game is the sum of the highest card ranks in each firework, for a highest possible score of 25.

More facts about Hanabi:

1. The Dec-POMDP symmetries correspond to permutations of the five card colors $(5! = 120)$.

2. In two-player Hanabi, there are 20 possible actions per turn, organized into four types: Play, Discard, Color Hint, and Rank Hint. These yield 190 distinct action transpositions.

3. A perfect score is 25, though some deck permutations make this score unreachable, so no policy can guarantee an expected return of 25.

