# OpenReview forum: "Expected Return Symmetries"
_ICLR.cc/2025/Conference — ICLR 2025 Poster_

### Official Review · Reviewer_4GFs · 2024-10-20

**Soundness:** 2
**Presentation:** 3
**Contribution:** 2
**Rating:** 6
**Confidence:** 3

**Summary:**

The paper generalizes the existing OP method to a more general symmetry group. They further define the group of expected return symmetries, and propose a novel method to learn expected return symmetries. The performance of the propose method significantly outperforms Dec-POMDP symmetries.

**Strengths:**

The introduction of expected return symmetries is novel, which is better suited for OP than Dec-POMDP symmetries. The proposed method greatly improves zero-shot coordination, significantly outperforming traditional Dec-POMDP symmetries. Moreover, expected return symmetries can be effectively applied in environments where agents act simultaneously.

**Weaknesses:**

1. The structure of the proposed expected return symmetries is quite similar to the state value funtion that evaluates the future expexted returns. It can be easily affected by the policies of other agents. In multi-agent settings, how the proposed method deals with non-stationary issue during self-play and cross play.
2. The authors argue that the optimal policies learned independently by an agent may not be compatible with other agents' policies during test time. It is not very clear that how the proposed method enhances the compatibility. How self-play optimal policies can differ significantly in coordination strategies.
3. The limitations of the proposed expected return symmetries should be discussed.

**Questions:**

See the weaknesses above.

---

> ### Author Response · Authors · 2024-11-19
> **Response to Reviewer 4GFs**
>
> We thank the reviewer for their thoughtful feedback and for recognizing the contributions of our work.
>
> ## 1. Dealing with Non-Stationarity:
>
> ### Comment:
> "The structure of the proposed expected return symmetries is quite similar to the state value function that evaluates the future expected returns. It can be easily affected by the policies of other agents. In multi-agent settings, how does the proposed method deal with non-stationary issues during self-play and cross-play?"
>
> ### Response:
> Expected return symmetries are, according to our definition, transformations that preserve the expected return of previously learned joint policies which are optimal in self-play. Therefore, the different self-play optimal joint policies are fixed and stationary during the learning of the expected return symmetries. Furthermore, since expected return symmetries are transforming joint policies, they take all of the different agents/local policies into account.
>
> ## 2. Enhancing Compatibility Between Agents:
>
> ### Comment:
> "It is not very clear how the proposed method enhances compatibility. How can self-play optimal policies differ significantly in coordination strategies?"
>
> ### Response:
> Thank you for the question. Self-play optimal policies can differ significantly because each independent training run may converge to different coordination conventions or strategies, especially in environments with multiple optimal solutions. In Hanabi, the incompatibility between such different conventions can be clearly seen, where the reported mean self-play score of IPPO policies is 24.04, whereas the mean XP score between different independently trained IPPO policies is 4.02. Another example is the two self-play optimal yet mutually incompatible policies that arise in the cat/dog game.
>
> Our method enhances compatibility across independent training runs, by forcing policies during training to be compatible with other policies within the same equivalence class. The policies in the equivalence class are constructed to also have a high self-play score, and therefore simulate the policies that can be converged upon in other training runs (i.e. simulate policies unseen rational partners might choose). The intuition here is a policy that is very compatible with all the policies in its equivalence class is also likely to be compatible with policies it would meet in cross-play. We have expanded the explanation in the paper to make this clearer.
>
> ## 3. Discussion of Limitations:
>
> ### Comment:
> "The limitations of the proposed expected return symmetries should be discussed."
>
> ### Response:
> We agree that discussing limitations is important. In response, we have expanded the conclusion to address the limitations of our approach. We discuss the reliance on access to optimal self-play policies, and the failure case where expected return symmetries aren’t sufficiently expressive to prevent coordination failure. We believe this addition provides a balanced view of our method and highlights areas for future research.
>
> Thank you again for your valuable feedback, which has helped us improve the clarity and completeness of our paper. Please let us know how we can make further improvements.

---

> > ### Comment · Reviewer_4GFs · 2024-11-25
> >
> > I appreciate the authors' efforts. My concerns have been majorly solved. I decide to increase my score.

---

> > > ### Author Response · Authors · 2024-11-25
> > > **Response to Reviewer 4GFs**
> > >
> > > We sincerely thank you for your thoughtful review and for acknowledging the improvements made to address your concerns. We appreciate the increase in your score and are glad to hear that the revisions have resolved the major issues you raised. Your feedback has been invaluable in strengthening the work.
> > >
> > > If there are any remaining points or suggestions for further refinement, we would be happy to address them.

---

### Official Review · Reviewer_GMBG · 2024-11-03

**Soundness:** 3
**Presentation:** 2
**Contribution:** 3
**Rating:** 5
**Confidence:** 3

**Summary:**

This paper extends the concept of Dec-POMDP symmetry to Expected Return (ER) symmetry for the Other-Play (OP) objective. ER symmetry captures broader classes of symmetry-breaking by preserving the expected return of self-play optimal policies instead of the state, action, and observation spaces, thus increasing diversity within equivalence classes at the cost of some optimality. The authors introduce novel algorithms to discover ER symmetries and validate their effectiveness across three MARL environments with independent agents, demonstrating improvement in terms of zero-shot coordination compared to existing symmetry discovery methods.

**Strengths:**

+ The introduction of ER symmetry seems to be a new contribution, extending the symmetry-breaking methods beyond simple action and observation relabeling.
+ The use of various toy examples throughout the paper makes the key definitions and concepts easy to understand.
+ The proposed approach is solid, and its effectiveness is underpinned by the with well-defined constructions of equivalence classes.

**Weaknesses:**

- Despite the use of the toy examples and the effort by the authors, I feel the writing could be further improved for readability. Some sections are still a bit challenging to follow.
- The experiments largely focus on the Hanabi environment, while the results on overcook and cat/dog environments are very brief. More  complex environments, especially those with continuous action and state spaces, should be used to demonstrate the effectiveness and scalability of the ER symmetry method, which unfortunately remain unclear.
- For Hanabi, the comparisons are limited to self-play and cross-play using IPPO. How do the proposed methods compare with (i) self- or cross-play using other RL algorithms, (ii) SOTA MARL baselines in Hanabi that do not necessarily employ self- or cross-play? I'm mostly interested in seeing how the proposed algorithms compare with SOTA MARL algorithms out there.

**Questions:**

1. Please address the questions raised in the weakness section, regarding experiments on more complex environments and comparison with SOTA MARL baselines.
2. In the caption of Figure 2, the mean SP score for the baseline population is listed as “162.33 ± 0.14,” while the graph displays a mean of 6.74. Is this a typo? I'd suggest the authors to thoroughly check all results presented in the paper.
3. The evaluation of group properties reveals a high relative reconstruction loss (18.4%), does this violation of a group's property of closure under composition impact the performance of symmetry discovery and MARL performance greatly?
4. For real-world applications, could you elaborate on the interpretability of ER symmetries? Would it be feasible to construct more intuitive symmetry structures based on those discovered by your model, and how does this compare to the interpretability of Dec-POMDP symmetry?
5. Regarding Equation 9, how large is the deviation from optimality in practice? Specifically, what is the gap between Equation 9 and Equation 10 in your experimental results? Are these deviations minor enough to be negligible?

---

> ### Author Response · Authors · 2024-11-19
> **Response to Reviewer GMBG**
>
> We thank the reviewer for their detailed feedback and for recognizing the strengths of our work.
>
> ## 1. Writing and Readability:
>
> ### Comment:
> "Despite the use of the toy examples and the effort by the authors, I feel the writing could be further improved for readability. Some sections are still a bit challenging to follow."
>
> ### Response:
> Thank you for highlighting this. We have revisited the manuscript to improve the clarity and readability of the sections that were challenging. Specifically, we have refined some of the explanations in Sections 3 and 4, ensuring that the key concepts are presented more intuitively. We have also added a second toy environment in Section 4.1, the iterated three-lever game, which we hope will improve intuition and understanding of our method.
>
> ## 2. Use of More Complex Environments:
>
> ### Comment:
> "More complex environments, especially those with continuous action and state spaces, should be used to demonstrate the effectiveness and scalability of the ER symmetry method, which unfortunately remain unclear."
>
> ### Response:
> We note that Hanabi is a challenging ZSC task, and there is ample precedent in the cooperation literature for papers to use it as the sole non-trivial environment on which they report results [1-6]. The original Overcooked is also often used as the sole benchmark for many works in the cooperation literature [7-10]. Furthermore, we emphasize that Overcooked V2 is a more challenging environment than the original Overcooked, as it limits agent observations within a configurable radius and introduces hidden information sources. Therefore, since we run experiments on not one but both of Hanabi and Overcooked V2, we maintain that our experimental exposition is strong relative to the existing cooperation literature, but are very open to specific suggestions the reviewer might have for suitable benchmarks.  We have edited the writing to highlight why we believe the chosen benchmarks are suitable for ZSC.
>
> ## 3. Comparison with Other RL Algorithms and SOTA MARL Baselines:
>
> ### Comment:
> "How do the proposed methods compare with (i) self- or cross-play using other RL algorithms, (ii) SOTA MARL baselines in Hanabi that do not necessarily employ self- or cross-play?"
>
> ### Response:
> (i) Comparison with other RL algorithms:
>
> Thank you for the suggested additional experiments. We have extended the comparison with Off-Belief Learning (OBL) [4] in our paper by adding another toy environment in Section 4.1, complete with experiments, where our method succeeds, but OBL completely fails: the iterated 3-lever game. We also note that the use of OBL in environments where agents act simultaneously is not straightforward: while any simultaneous action game can be converted into a turn-based one, the resulting turn-based game is not unique, and OBL might give different policies for different choices of the turn-based game [4]. Therefore, OBL is less applicable for environments such as in Overcooked V2, whereas our method remains effective. As well, we compare our method’s ability to find grounded communication protocols with cognitive hierarchy-based approaches [11-12].
>
> (ii) Comparison with SOTA MARL baselines:
>
> We understand the interest in comparing with other MARL algorithms. In Hanabi, most state-of-the-art methods focus on self-play (SP) performance, which does not necessarily translate to zero-shot coordination capabilities. Our focus is on improving cross-play (XP) performance without sacrificing SP performance significantly. We have included comparisons with Independent PPO (IPPO), which is a strong baseline for SP in Hanabi. Since our method aims to enhance coordination between independently trained agents, we believe this comparison is appropriate. We are open to including additional baselines if specific methods are suggested.
>
> ## 4. Figure 2 Clarification
>
> ### Comment:
> "In the caption of Figure 2, the mean SP score for the baseline population is listed as '162.33 ± 0.14,' while the graph displays a mean of 6.74. Is this a typo?"
>
> ### Response:
> Thank you for pointing this out, we have added clarifying text in the paper to prevent misunderstanding. This is not a typo, we apologize for any confusion. The mean score of 162.33 ± 0.14 refers to the self-play performance of the SP-optimal IPPO baseline population, whereas the graph displays the cross-play scores of IPPO, with a mean of 6.74. This illustrates the gap between SP and XP performance. We see that the use of ER symmetries results in a mean self-play performance of 27.81 and a mean cross-play performance of 15.80, showing that ER symmetries effectively reduce the gap between SP and XP.

---

> > ### Comment · Reviewer_GMBG · 2024-11-25
> >
> > Thank you the response to my questions. I still feel add more baselines (even if these are self-play like authors mentioned) and comparisons in more complex tasks are needed, and would help understand the benefits (and potential limitations) of the proposed method in practice. I maintain my opinion that the paper falls somewhere between 5 and 6.

---

> ### Author Response · Authors · 2024-11-19
> **Response to Reviewer GMBG [continued]**
>
> ## 5. Impact of Violation of Group Properties:
>
> ### Comment:
> "The evaluation of group properties reveals a high relative reconstruction loss (18.4%), does this violation of a group's property of closure under composition impact the performance of symmetry discovery and MARL performance greatly?"
>
> ### Response:
> Indeed, the learned symmetries are approximate, and the reconstruction loss indicates deviations from perfect group properties. However, our experiments show that even with this approximation, the learned symmetries significantly improve coordination performance. Regularization helps to significantly reduce the reconstruction loss, enhancing the effectiveness of the symmetries. We have added discussion in the paper to address this point and acknowledge that future work could focus on further reducing this loss. Note we have moved this table to the appendix to make space for more experiments, such as the new iterated 3-lever game.
>
> ## 6. Interpretability of ER Symmetries:
> ### Comment:
> "For real-world applications, could you elaborate on the interpretability of ER symmetries? Would it be feasible to construct more intuitive symmetry structures based on those discovered by your model, and how does this compare to the interpretability of Dec-POMDP symmetry?"
>
> ### Response:
> We appreciate this question. Firstly, interpretability and transparency in conventions is paramount for success in coordination. Since we demonstrate ER symmetries can guide the design of policies that are more compatible with a variety of partners, they thus improve interpretability of the trained policies. In light of your comments, we have added experiments that compare the conditional action matrices of the OP+ER symmetry agents with the OP+Dec-POMDP symmetry agents to the appendix, showing how the ER symmetries lead to agents with more interpretable play.
>
> ## 7. Deviation from Optimality in Practice:
>
> ### Comment:
> "Regarding Equation 9, how large is the deviation from optimality in practice? Are these deviations minor enough to be negligible?"
>
> ### Response:
> Since using these symmetries still significantly improves ZSC in all the considered environments, practically speaking the deviation is minor with respect to our ultimate goal. We have clarified this point in the paper to emphasize that the trade-off is acceptable for achieving better ZSC.
>
> Thank you again for your comprehensive feedback. We believe that addressing these points has strengthened the paper, and we are eager to hear how else we can further improve the paper.
>
> [1] A. Lupu, B. Cui, H. Hu, and J. Foerster. Trajectory diversity for zero-shot coordination. In the International Conference on Machine Learning, pages 7204–7213. PMLR, 2021.
>
> [2] Siu, H. C., Peña, J., Chen, E., Zhou, Y., Lopez, V., Palko, K., ... & Allen, R. (2021). Evaluation of human-AI teams for learned and rule-based agents in Hanabi. Advances in Neural Information Processing Systems, 34, 16183-16195.
>
> [3] H. Hu, A. Lerer, A. Peysakhovich, and J. Foerster. “Other-play” for zero-shot coordination. In H. D. III and A. Singh, editors, Proceedings of the 37th International Conference on Machine Learning, volume 119 of Proceedings of Machine Learning Research, pages 4399–4410. PMLR, 13–18 Jul 2020.
>
> [4] H. Hu, A. Lerer, B. Cui, L. Pineda, D. Wu, N. Brown, and J. N. Foerster. Off-belief learning. ICML, 2021. URL https://arxiv.org/abs/2103.04000.
>
> [5] Nekoei, H., Badrinaaraayanan, A., Courville, A., & Chandar, S. (2021, July). Continuous coordination as a realistic scenario for lifelong learning. In the International Conference on Machine Learning (pp. 8016-8024). PMLR.
>
> [6] Canaan, R., Togelius, J., Nealen, A., & Menzel, S. (2019, August). Diverse agents for ad-hoc cooperation in hanabi. In 2019 IEEE Conference on Games (CoG) (pp. 1-8). IEEE.
>
> [7] Micah Carroll, Rohin Shah, Mark K Ho, Tom Griffiths, Sanjit Seshia, Pieter Abbeel, and Anca Dragan. On the utility of learning about humans for human-AI coordination. In Advances in Neural Information Processing Systems, pages 5175–5186, 2019.
>
> [8] Strouse, D. J., et al. "Collaborating with humans without human data." Advances in Neural Information Processing Systems 34 (2021): 14502-14515.
>
> [9] R. Charakorn, P. Manoonpong, and N. Dilokthanakul. Investigating partner diversification methods in cooperative multi-agent deep reinforcement learning. In International Conference on Neural Information Processing, 2020.
>
> [10] P. Knott, M. Carroll, S. Devlin, K. Ciosek, K. Hofmann, A. Dragan, and R. Shah. Evaluating the robustness of collaborative agents. In International Conference on Autonomous Agents and Multiagent Systems (AAMAS), 2021.
>
> [11] Brandon Cui, Hengyuan Hu, Luis Pineda, and Jakob Foerster. K-level reasoning for zero-shot
> coordination in hanabi. Advances in Neural Information Processing Systems,
> 2021.
>
> [12] Colin F Camerer, Teck-Hua Ho, and Juin-Kuan Chong. A cognitive hierarchy model of games. The
> Quarterly Journal of Economics, 2004.

---

> ### Author Response · Authors · 2024-11-25
> **Response to Reviewer GMBG**
>
> Thank you for your feedback and for taking the time to review our responses.
>
> To address your comments, we have included discussion of the two-lever variant of the iterated three-lever game in the revised manuscript. This variant highlights a notable limitation of our method, which we believe contributes to a better understanding of its benefits and constraints. We hope this addition provides further clarity and addresses the concerns raised about demonstrating the practical implications of our approach.
>
> We are happy to provide additional comparisons if the reviewer or the community can suggest specific baselines or benchmarks. In the absence of such suggestions, we have expanded our evaluation to the best of our ability within the scope of this work. To reemphasize a point in our previous response, existing literature on coordination typically just takes one of Hanabi or Overcooked as the single complex environment considered [1-10]. Thus, by including not just Hanabi, but also Overcooked V2, a more complex version of the original Overcooked, we maintain that our experimental exposition is strong compared to existing literature on coordination.
>
> We hope that these efforts reflect our commitment to improving the manuscript and engaging constructively with the reviewer’s comments. We respectfully ask the AC to consider the work done to address these concerns and note that our revisions aim to provide a fair and comprehensive evaluation of our method.
>
> [1] A. Lupu, B. Cui, H. Hu, and J. Foerster. Trajectory diversity for zero-shot coordination. In the International Conference on Machine Learning, pages 7204–7213. PMLR, 2021.
>
> [2] Siu, H. C., Peña, J., Chen, E., Zhou, Y., Lopez, V., Palko, K., ... & Allen, R. (2021). Evaluation of human-AI teams for learned and rule-based agents in Hanabi. Advances in Neural Information Processing Systems, 34, 16183-16195.
>
> [3] H. Hu, A. Lerer, A. Peysakhovich, and J. Foerster. “Other-play” for zero-shot coordination. In H. D. III and A. Singh, editors, Proceedings of the 37th International Conference on Machine Learning, volume 119 of Proceedings of Machine Learning Research, pages 4399–4410. PMLR, 13–18 Jul 2020.
>
> [4] H. Hu, A. Lerer, B. Cui, L. Pineda, D. Wu, N. Brown, and J. N. Foerster. Off-belief learning. ICML, 2021. URL https://arxiv.org/abs/2103.04000.
>
> [5] Nekoei, H., Badrinaaraayanan, A., Courville, A., & Chandar, S. (2021, July). Continuous coordination as a realistic scenario for lifelong learning. In the International Conference on Machine Learning (pp. 8016-8024). PMLR.
>
> [6] Canaan, R., Togelius, J., Nealen, A., & Menzel, S. (2019, August). Diverse agents for ad-hoc cooperation in hanabi. In 2019 IEEE Conference on Games (CoG) (pp. 1-8). IEEE.
>
> [7] Micah Carroll, Rohin Shah, Mark K Ho, Tom Griffiths, Sanjit Seshia, Pieter Abbeel, and Anca Dragan. On the utility of learning about humans for human-AI coordination. In Advances in Neural Information Processing Systems, pages 5175–5186, 2019.
>
> [8] Strouse, D. J., et al. "Collaborating with humans without human data." Advances in Neural Information Processing Systems 34 (2021): 14502-14515.
>
> [9] R. Charakorn, P. Manoonpong, and N. Dilokthanakul. Investigating partner diversification methods in cooperative multi-agent deep reinforcement learning. In International Conference on Neural Information Processing, 2020.
>
> [10] P. Knott, M. Carroll, S. Devlin, K. Ciosek, K. Hofmann, A. Dragan, and R. Shah. Evaluating the robustness of collaborative agents. In International Conference on Autonomous Agents and Multiagent Systems (AAMAS), 2021.

---

### Official Review · Reviewer_zi1H · 2024-11-04

**Soundness:** 4
**Presentation:** 3
**Contribution:** 3
**Rating:** 8
**Confidence:** 3

**Summary:**

This work explores *symmetry in expected returns* as an inductive bias for training agents that are more capable of zero-shot coordination.

**Strengths:**

- The writing is clear, concise, and detailed.
- The approach is well-motivated theoretically, and seemingly well-grounded in prior literature.
- Discussion of prior work appears complete.
- The experiments are compelling, and validate the claims made in the paper.
- Future directions are compelling.
- Work is likely to be of interest to the broader field.

**Weaknesses:**

- Limitations are not adequately addressed. I would suggested addressing this explicitly in the conclusion, between the summary and discussion of future directions.
- Readability:
	- Figure 2 graph text is far too small.
	- What purpose do the black horizontal bars at the top and bottom of Figure 1's image serve? Seems the text would fit much better without these.

**Questions:**

- What are the key limitations of this approach?

---

> ### Author Response · Authors · 2024-11-19
> **Response to Reviewer zi1H**
>
> We thank the reviewer for their thoughtful feedback and positive assessment of our work.
>
> ## Comment:
> "Limitations are not adequately addressed. I would suggest addressing this explicitly in the conclusion, between the summary and discussion of future directions."
>
> ## Response:
> We appreciate this suggestion. In response, we have added a dedicated paragraph in the conclusion explicitly discussing the limitations of our approach. Specifically, we comment on potential limits with the expressivity of the considered transformation class. Additionally, our method has tacitly assumed that near-optimal policies are accessible; in settings where we learn symmetries over significantly sub-optimal self-play policies, the effectiveness of our outlined approach remains unclear. We believe this addition provides a clearer understanding of the limitations and potential areas for future work.
>
> ### Readability Concerns:
> We have adjusted Figure 1 to remove unnecessary black bars and ensure the text fits better, and have increased the font size in Figure 2 to enhance readability.
>
> Thank you again for your constructive feedback, which has helped us improve the clarity and completeness of our paper.

---

> > ### Comment · Reviewer_zi1H · 2024-11-24
> >
> > I thank the authors for addressing the concerns raised. I maintain a positive opinion of the paper and will keep my initial score.

---

> > > ### Author Response · Authors · 2024-11-25
> > > **Response to Reviewer zi1H**
> > >
> > > We sincerely thank the reviewer for their follow-up and for maintaining a positive opinion of our work. We appreciate the opportunity to address your concerns. We are glad the revisions met your expectations, and we are grateful for your thorough evaluation and constructive comments throughout this process.

---

### Official Review · Reviewer_8d16 · 2024-11-04

**Soundness:** 3
**Presentation:** 2
**Contribution:** 3
**Rating:** 6
**Confidence:** 1

**Summary:**

This paper addresses the zero-shot coordination problem in multi-agent systems within the context of decentralized partially observable Markov decision processes. It introduces a new symmetric class, which is a class containing the previously proposed environment symmetries. The paper also presents an algorithm to find these symmetries and demonstrates that agents within this symmetric class can achieve better zero-shot coordination in decentralized partially observable Markov decision processes.

**Strengths:**

1. The organization of this paper is well-structured.
2. The paper conducts experiments in several zero-shot coordination tasks and gives comprehensive analysis.

**Weaknesses:**

As I'm not familiar with this field, it is hard for me to give constructive suggestions to the authors.

**Questions:**

As my understanding of this paper is limited, I hope the authors can provide an intuitive example to illustrate why their proposed symmetry class is better than the Other-Play algorithm [1].

[1] Hu, Hengyuan, et al. "“other-play” for zero-shot coordination." International Conference on Machine Learning. PMLR, 2020.

---

> ### Author Response · Authors · 2024-11-19
> **Response to Reviewer 8d16**
>
> We thank the reviewer for their time and for their positive assessment of our work.
>
> # Question:
> "As my understanding of this paper is limited, I hope the authors can provide an intuitive example to illustrate why their proposed symmetry class is better than the Other-Play algorithm [1]."
>
> # Response:
>
> Certainly, we appreciate the opportunity to clarify this point. There are two advantages of expected return symmetries over the Dec-POMDP symmetries used in the original Other-Play paper.
>
> ## Expected Return Symmetries leverage unknown symmetries
> First, to apply the Other-Play algorithm with Dec-POMDP symmetries one needs to know the Dec-POMDP symmetries beforehand. This is rarely the case for complex environments, where Dec-POMDP symmetries aren’t as obvious as the color switches in Hanabi. Thus any symmetries to be used in the other-play algorithm first need to be discovered, and our algorithm is able to do so, not requiring any a priori knowledge about symmetries in the Dec-POMDP.
>
> ## Expected Return Symmetries leverage a broader class of symmetries
> Second, the class of expected return symmetries is broader, and much more flexible, than the class of Dec-POMDP symmetries, and is thus able to make policies symmetric to each other that would not be symmetric to each other w.r.t. the class of Dec-POMDP symmetries. This increased flexibility, or expressiveness, then leads to more diversity within equivalence classes, which then leads to policies that are optimal w.r.t. the other-play objective using expected return symmetries to be compatible with a much broader set of policies.
> As an example for this increased expressiveness and flexibility, consider the "cat/dog" coordination game we describe in the paper. Two agents independently choose between "cat" and "dog" and receive a reward if they choose the same animal. There is a lightbulb that can be "on" or "off," providing an observation to the agents. In the original Other-Play (OP) algorithm [1], which relies on Dec-POMDP symmetries, action and observation relabelings that preserve the environment dynamics are used to improve coordination. However, in this game, such symmetries are not present because swapping "cat" with "dog" or "on" with "off" individually does not preserve the game's dynamics (they are each associated with different rewards).
>
> Therefore, in this cat/dog game, the two incompatible self-play optimal policies are not symmetric w.r.t. the Dec-POMDP symmetries used by the original other-play algorithm, but they are symmetric w.r.t. expected return symmetries. The other-play learning rule using expected return symmetries yields the grounded policy, which leads to better XP scores than self-play. Since the only Dec-POMDP symmetry in this cat/dog game is the identity transformation, the original other-play algorithm does not help us and becomes equivalent to self-play, because it does not consider the two optimal self-play policies as symmetric to each other. Since self-play does not take into account compatibility when converging onto a policy, self-play will converge onto either of them with roughly equal probability, which leads to coordination failure at test-time.
>
> Thus our proposed expected return symmetries identify a broader symmetry class than Dec-POMDPs, which allows agents to coordinate more effectively without prior agreement, addressing coordination challenges that Dec-POMDP symmetries used in [1] cannot resolve due to their reliance on stricter symmetry constraints.
>
> [1] H. Hu, A. Lerer, A. Peysakhovich, and J. Foerster. “Other-play” for zero-shot coordination. In H. D. III and A. Singh, editors, Proceedings of the 37th International Conference on Machine Learning, volume 119 of Proceedings of Machine Learning Research, pages 4399–4410. PMLR, 13–18 Jul 2020.

---

> > ### Comment · Reviewer_8d16 · 2024-11-22
> >
> > I thank the authors for their detailed explanation to my questions and my concern is well addressed. Based on the overall quality of this paper, I decide to remain my score.

---

> > > ### Author Response · Authors · 2024-11-22
> > > **Reply to Reviewer 8d16**
> > >
> > > We sincerely thank you for your engagement with our work and for confirming that your concerns have been well addressed.
> > >
> > > Given that your initial concerns have been resolved, we kindly ask if you might reconsider your score to better reflect the strengths of the paper, such as its contributions to advancing zero-shot coordination and its comprehensive analysis across multiple tasks. However, if there are additional concerns, questions, or suggestions for improvement that we may have overlooked, we would be very grateful to hear them and further refine the manuscript.
> > >
> > > Thank you again for your valuable feedback and support in improving the quality of this work.

---

### Meta-Review · Area_Chair_871P · 2024-12-22

**Metareview:**

This paper introduce a new family of environment symmetry in partially observable MARL called the Expected Return Symmetries (ERS), which generalizes over previous known symmetry classes. It is then shown that agents trained to be compatible under the group of ERS achieve better zero-shot coordination results than those using environment symmetries. This is overall an interesting paper on an under-explored topic in RL. With the fast developments of AI agent, MARL interaction will inevitably become more and more common, and therefore this is a very timely problem to study.

**Additional Comments On Reviewer Discussion:**

NA

---

### Decision · Program_Chairs · 2025-01-22

Accept (Poster)